# The Effects of Vegetation Structure and Timber Harvesting on Ground Beetle (Col.: Carabidae) and Arachnid Communities (Arach.: Araneae, Opiliones) in Short-Rotation Coppices

Jessika Konrad * , Ralph Platen and Michael Glemnitz

Research Area 2—Land Use and Governance, Leibniz-Center for Agricultural Landscape Research (ZALF), Eberswalder Straße 84, 15374 Müncheberg, Germany
* Correspondence: konrad@zalf.de

**Abstract:** Landscape complexity is a crucial factor for insect diversity in agricultural landscapes. Short-rotation coppices (SRCs) are characterised by high habitat heterogeneity. The impact of vegetation structure on the composition and diversity of ground beetle and arachnid communities was therefore investigated in four SRCs and six reference plots. The study site was located in Hesse, Germany. The invertebrates were surveyed from 2011 to 2014 using pitfall traps, and the vegetation structure was quantified by estimating the percentage cover of 10 structural variables. The impact of the selected structural variables on community composition was analysed during grove growth as well as after a timber harvest. We found correlations between the cover percentages of structural variables and the quantitative and qualitative species composition in both animal groups ($p \leq 0.05$). The share of individuals of forest species increased with rising shading and litter cover, while those of open land decreased. The opposite trends were found the year after the timber harvest. The SRCs showed a higher structural diversity compared to the reference biotopes ($p \leq 0.05$). This was positively correlated ($p \leq 0.001$) with species diversity and the variety of habitat preference groups in both animal groups. The high diversity within the habitat preference groups indicated a functional redundancy among species for both animal groups and, consequently, a high level of resilience within these communities. Little is known about the functional aspects of ground beetles and spiders in ecosystems, and detailed studies are urgently needed. We conclude that SRCs can contribute to the diversification of agricultural landscapes as an alternative to traditional crop cultivation.

**Keywords:** agroforestry; arthropods; biodiversity; energy crops; fast-growing trees; heterogeneity; land use change





## 1. Introduction

Numerous studies have addressed the changes in insect diversity within the agricultural landscape and their causes. Land use change [1–4], agricultural intensification [2,5], high levels of pesticide application [6,7], narrow crop rotations [8,9], and the absence of landscape complexity [10–12] are all considered major contributors to the decline in insect diversity. The fact that more than half of Germany's total land area was used for agriculture in 2022 [13] illustrates the relevance of agriculture for the conservation of biodiversity. However, as insect declines are not limited to agricultural areas but have also been observed in forests [14–17], other influencing factors such as climate change are increasingly being discussed. Sánchez-Bayo and Wyckhuys [6] emphasise that in tropical regions, climate change is an important driver of insect decline, whereas in temperate zones, it is confined to mountainous regions. Outhwaite et al. [4] attribute this to the narrower thermal tolerances of tropical species. Other studies suggest that recent warming, associated with higher local temperatures, should at least promote insect biomass and species richness within temperate climate regions [3,4,18]. Yet, a 10-year study carried out around 100 km away from the study area confirms a decline in insects that is primarily related to management

intensity [19]. The authors highlight that, in particular, predators and decomposers of deadwood are adversely affected in highly managed pine and spruce monocultures with low structural complexity.

In general, the loss of suitable habitats is a major driver of species decline. Especially for species that are reliant on various habitats during their life cycle, the close proximity and accessibility to different habitats for development, foraging, hibernation, and reproduction are crucial [11]. In homogeneous structurally impoverished landscapes, these habitats are often not reachable within traversable distances. Moreover, different groups of organisms have varying habitat requirements. For example, ground beetles require sufficient ground rest for undisturbed larval development, which is important for imago and larval hibernators at different times of the year [20]. Pollinators like bumblebees (*Bombus terrestris*) depend on sufficient floral resources for foraging during the growing season [21], while web-spinning spiders inhabit structurally rich habitats where they find anchoring points for web construction [22]. The diverse habitat needs of various organism groups necessitate a complex habitat array within the agricultural landscape. Tscharntke et al. [23], therefore, include the spatial and temporal diversification of cultivated areas, including the cultivation of catch crops or agroforestry, among the measures to promote biodiversity. Another aspect involves landscape fragmentation and the resulting degree of isolation among habitats [24]. When local extinction rates, for instance, in forests due to isolation, exceed immigration rates, this leads to the extinction of local populations [25]. For less mobile species like flightless ground beetles, the large distances between forests often present insurmountable barriers [26]. Diverse landscapes, where wooded habitats act as connecting elements, allowing the exchange between distant forests, can contribute to preserving insect diversity. Veste and Böhm [27] emphasise the potential of a habitat connectivity function in SRC cultivation.

Numerous studies conducted over the past 20 years have aimed to answer the question of how the establishment of SRCs impacts the surrounding agricultural landscape and its contribution to enhancing biodiversity within the agricultural environment [27–29]. These studies included vegetation analyses [30] and bird assessments [31] but primarily focused on invertebrates [29,32]. Ground beetles were the most frequently studied, followed by bird investigations [33]. The shortcomings of these studies often involved limited study durations (typically less than a year), a restricted sample selection, and insufficient replication, leading only to very limited, locally applicable conclusions [28]. Furthermore, the climatic and petrographic disparities between study regions did not permit generalised statements about ground beetle communities [32]. Long-term studies on the influence of the stand age of SRCs are very rare due to the short time since the introduction of this form of agricultural practice, as well as constraints in financial and human resources [32].

Ground beetles are among the most diverse beetle families worldwide, known for their well-documented biology and ecology and due to their predominantly epigean lifestyle, allowing for a comprehensive quantitative assessment [34]. Alongside ground beetles, arachnids are quantitatively significant regulators of animal pests in the agricultural landscape [35]. Like ground beetles, the majority of species have an epigean lifestyle. However, since most spiders can construct webs, their presence and abundance are more significantly dependent on vegetation structure compared to ground beetles [36]. Tews et al. [37] suggest that structurally complex habitats offer more niches and diverse opportunities for utilising environmental resources. Bianchi et al. [38] also highlight the ecological significance of heterogeneously structured agricultural landscapes in the natural pest regulation by ground beetles and spiders. As demonstrated by Ribera et al. [39], with ground beetles, their species-specific traits are statistically highly correlated with the habitat characteristics. An increase in landscape structural diversity generally predicts a positive effect on species diversity [40,41].

The present study is based on a four-year investigation conducted across four short-rotation coppices (SRCs) at varying stages of growth and six reference plots in the surrounding agricultural landscape. The overarching objective was to analyse the impact of vegetation structure on the composition of ground beetle and arachnid communities within SRCs in comparison to reference plots. SRCs are characterised by low management

intensity, reduced pesticide use, and long-term soil dormancy. Therefore, SRCs can potentially contribute to the maintenance of ground beetle and arachnid species diversity in agricultural landscapes. One aim of this study was to clarify whether SRCs have higher species numbers and diversities than the reference biotopes. Compared to traditional agricultural biotopes, SRCs are characterised by a higher structural diversity. We investigated whether this habitat heterogeneity in SRCs leads to a higher number of species with different habitat preferences compared to reference biotopes. During the growth phase of groves, habitat characteristics, such as ground vegetation, litter cover, and the extent of shading, change fundamentally. For this reason, changes in the qualitative and quantitative composition of ground beetle and arachnid communities and their ecological characteristics were analysed over time. We analysed whether forest species show higher proportions of individuals during the growth phase in each year of regrowth with respectively lower shares of open-land species than in the previous year. Timber harvesting implies abrupt and comprehensive changes in the structural characteristics of these habitats. For both animal groups, we determined whether communities differ in the proportion of individuals of forest and open-land species in the year after timber harvesting compared to the previous year. The following hypotheses were formulated: (i) The composition and diversity of ground beetle and arachnid cenoses differ statistically significantly over time and spatially between the SRCs compared to the reference plots. (ii) The ecological traits of ground beetles and arachnids are correlated with the extent of coverage provided by selected structural variables of the vegetation. (iii) Changes in the vegetation structure and the composition of ground beetle and arachnid cenoses occur over the course of the growth phase and after timber harvesting of the SRCs, and these changes are correlated with each other.

## 2. Materials and Methods

### 2.1. Study Area

The study area, Haine (WGS84: 51.041335° N, 08.720965° E), is located east of the city of Allendorf/Eder in the state of Hesse in Germany, at an altitude of 299 metres above sea level, transitioning from the Rhenish Slate Mountains to the Hessian Highlands (Figure 1). The predominant soil types in this region, abundant in grassland and beech forests, are brown soils, which have a medium yield potential of 29 points [42].

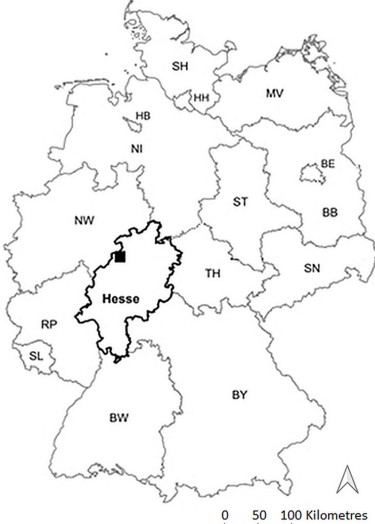

**Figure 1.** Location of the Haine study area (square) in the federal state of Hesse. Map of the Federal Republic of Germany with the individual federal states marked at a scale of 1:250,000. Adapted with permission from [43], 2011, © GeoBasis-DE/BKG, modified. SH = Schleswig–Holstein, HH = Hamburg, HB = Bremen, NI = Lower Saxony, MV = Mecklenburg–Western Pomerania, BE = Berlin, BB = Brandenburg, ST = Saxony–Anhalt, NW = North Rhine–Westphalia, SN = Saxony, TH = Thuringia, RP = Rhineland–Palatinate, SL = Saarland, BY = Bavaria, and BW = Baden–Württemberg.

The area comprises 68,039 ha of agricultural land, divided into 58% arable land and 42% grassland [44]. A total area of 302 ha (0.4%) is used for SRCs [45]. The average annual temperature at the Burgwald-Bottendorf climate station, located about seven kilometres from the study area, is 8.4 °C, and the average annual precipitation is 689.8 mm (30-year mean 1985–2014) [46].

## 2.2. Experimental Design

The study comprised four short-rotation coppices (SRCs) and six reference plots (Figure 2). SRC1, SRC3, and SRC4 were planted with poplar hybrids of the varieties Max 1–Max 4 (*Populus maximowiczii* U. Thobae × *Populus nigra* L. cr. Max 1–Max 4), while SRC2 was planted with the variety Muhle Larsen (*Populus nigra* L. × *Populus maximowiczii* A. Henry, NM6) on previously cultivated arable land. Timber harvesting was conducted in a rotation cycle every two to four years, resulting in up to four growth stages across the total area. SRC1 was not harvested during the study period and was in its fourth growth stage in 2014. SRC2 was harvested in the winter of 2013/2014 in its fourth growth stage, while SRC3 and SRC4 were harvested in the winter of 2012 in their third growth stage.

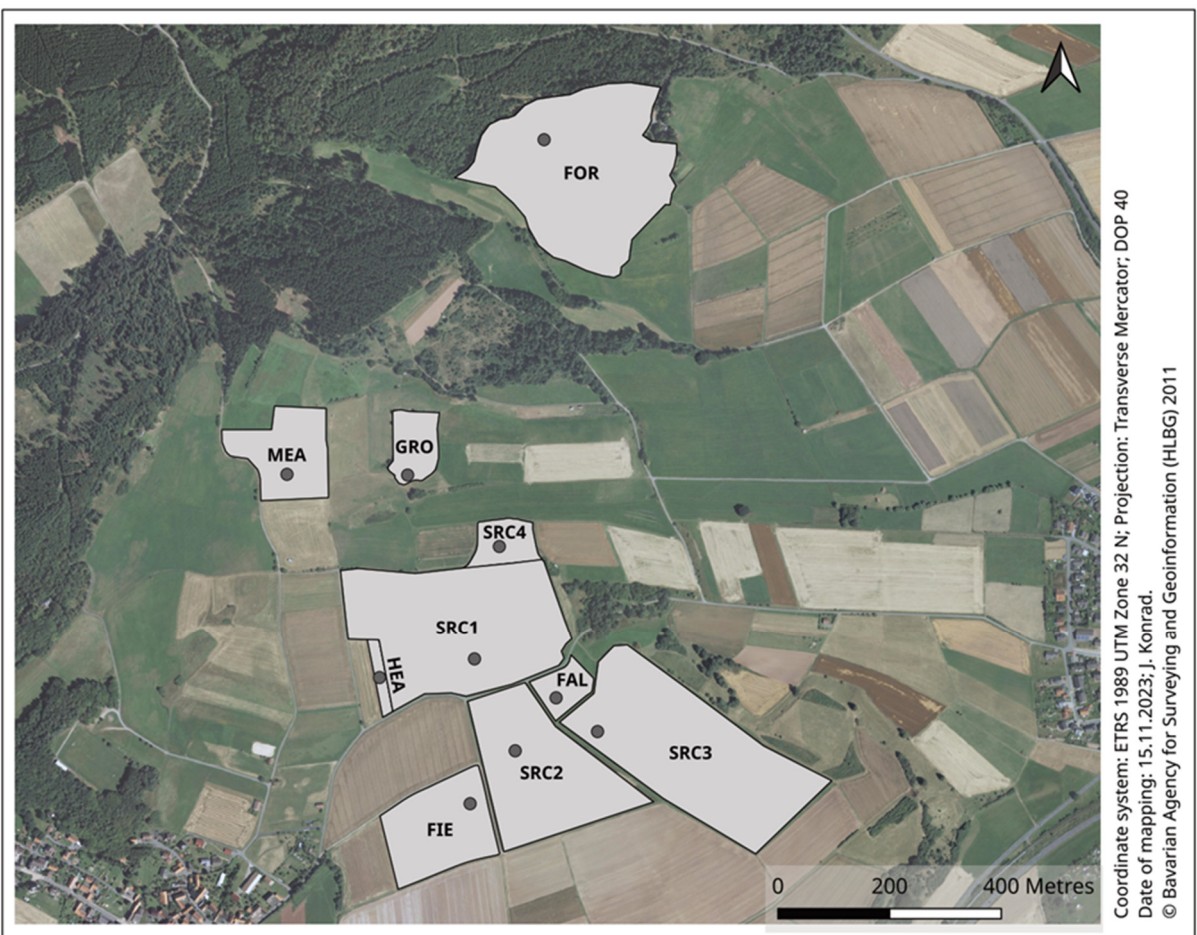

**Figure 2.** Location of the study plots and position of the pitfall traps (one trap point represents five pitfall traps in a linear transect). FIE = arable field, FAL = fallow, HEA = headland, MEA = meadow, GRO = grove, SRC1–SRC4 = short-rotation coppices, and FOR = forest.

As the interpretation of the results can also be influenced by the choice of reference biotopes, a total of six different biotope structures typical of the region were selected as reference plots. The arable field (FIE) reference plot was conventionally cultivated and planted with winter cereals (2011–2013) and summer cereals (2014). The fallow (FAL) was mulched annually in autumn, and in 2014, it was additionally mown twice. The grove

(GRO) was an isolated stand of trees, surrounded by grassland, consisting of beeches, oaks, and hazels. In the headland (HEA), directly adjacent to SRC1, a perennial wildflower mix was sown in 2011. The approximately 150-year-old oak–beech forest (FOR) had a loose shrub cover in addition to hornbeam. The meadow (MEA) was mown at the beginning of June and partially grazed by cattle until the end of October.

A vegetation structure assessment was conducted according to Dierschke [47] using 10 squares of 1 m² each, set up at a distance of 1 m to the right and left of the pitfall traps. Visual estimates were used to determine the percentage cover of shading by trees (shade), shrubs, crops, grass, herbaceous plants, moss, and litter. In addition, the percentage of deadwood ($\geq 2$ cm to $\leq 10$ cm and $\geq 10$ cm in diameter) and the extent of the vegetation-free, open ground were quantified. Except for calculating the structural diversity, the statistical analyses only included the variables of shade, herb, grass, litter, and open ground.

Ground beetles and arachnids were captured using five pitfall traps (6 cm in diameter) following Barber [48]. These traps, roofed with transparent plastic panels, were buried into the ground along a linear transect, spaced 5 metres apart. A 10% sodium benzoate solution was used as a non-attracting and human non-toxic capture and preservation liquid. This was mixed with approx. 20 mL acetic acid and a detergent. To obtain a detailed record of the composition of the ground beetle and arachnid cenoses, the surveys were carried out between 2011 and 2014 from the beginning of April to the end of October. The traps were changed every 14 days to ensure sufficient conservation in the summer months and to minimise losses due to wild animals. Their content was pre-sorted according to ground beetles and arachnids. The specimens were preserved in 70% ethanol until species identification.

### 2.3. Taxonomy and Nomenclature

The ground beetles were identified following Müller-Motzfeld [49], while the spiders were identified based on refs. [50–60], and harvestmen were identified following Martens [61]. The nomenclature for ground beetles followed Schmidt et al. [62], spiders followed Platnick [63], and harvestmen followed Muster et al. [64]. Due to their similar lifestyle, spiders and harvestmen were combined and analysed as arachnids. Data management was conducted using the database software MS Access from the Microsoft Office 2016 suite.

### 2.4. Ecological Groups
2.4.1. Ecological Types (ETs)

Each ground beetle and arachnid species was assigned an ecological trait that characterises the species' preference for abiotic factors such as moisture and light in the field [65,66] (Tables S1 and S2). These individual traits were classified into species-specific (differentiated) ecological types (ETs) (Table S3) and adapted for the broader geographic area of the western Central German Uplands [67]. For statistical analysis, the differentiated ecological types were aggregated into larger groups for each animal group (Table S4).

2.4.2. Habitat Preferences (HPs)

The habitat preferences (HPs) describe the habitat preferred by a species. For ground beetles, they were derived from the catalogue of the Society for Applied Carabidology for the western Central German Uplands [68]. For arachnid species, the HPs were adapted for the study area by Konrad [67] based on the information from Platen et al. [66]. The species-specific (differentiated) habitat preferences can be found in the species lists in the Appendices (Tables S1 and S2), with explanations provided in Table S5. The differentiated habitat preferences were summarised into six classes per animal group (Table S6) and, except for calculations concerning habitat preference diversities, were used for all statistical analyses.

### 2.5. Species, Habitat Preference, and Vegetation Structure Diversity

To characterise the alpha diversity within the ground beetle and arachnid cenoses, we used the number of species and the count of habitat preferences, and utilised the following diversity indices: log series $\alpha$, Shannon, Shannon exponential, reciprocal Simpson, and

evenness for species diversity [69] and habitat preference diversity (Tables S14–S17) [70]. The log series α emphasises species with very few individuals, the Shannon index highlights population variability, evenness signifies the equitable distribution of individuals among species, and the reciprocal Simpson index indicates dominant species.

For the calculation of vegetation structure diversity [67], we utilised the count and coverage of structural elements for the aforementioned indices. To compute the habitat preference diversity, we used the number of species-specific habitat preferences and their individual counts (Table S13). To compare the vegetation structure diversity with the species and habitat preference diversity among the study plots and individual years, the Shannon exponential index was employed [69,71].

### 2.6. Statistical Analysis

Individuals of species were normalised as per their aggregated ecological type and habitat preference group and were then subjected to an ArcSin square root transformation. The proportions of individuals for stenotopic and eurytopic forest species were Log10-transformed, while the species counts and diversity indices were analysed without transformation. Prior to the analyses, the data were checked for normal distribution using the Shapiro–Wilk test. The equality of variances among the plots was assessed via the Levene test, and the equality of means was verified with the Welch and Brown–Forsythe tests. Outliers were mostly retained in the dataset. The analyses were conducted based on single traps, except for those involving RDA, the comparison of plots using Shannon exponential indices, and the simple regression between vegetation structure, species, and habitat preference diversity. For these, individual counts from the five single traps within a plot were averaged. Differences in ecological types, habitat preferences, and diversity indices among the plots and study years were analysed using one-way ANOVA, and statistical significance was assessed using the Duncan test ($p \leq 0.05$).

To test the hypothesis that there is a relationship between the quantitative and qualitative composition of ground beetle and arachnid communities and the vegetation structure variables in the study plots, redundancy analyses (RDAs) were conducted. The results, along with five structural variables, were graphically represented in ordination diagrams for the first and second ordination axes. For an interpretation of the ordination diagrams, refer to Šmilauer and Lepš [72]. The vegetation structure variables of shade, herb, grass, litter layer, and open ground were included in the analyses. To assess the influence of vegetation structure in individual growth years on the community composition, the SRC plots were differentiated based on growth years and jointly analysed with the respective reference plots examined during the same period. Due to the inhomogeneous number of SRCs of the same growth years, the analyses included 22 plots of cenoses in the first two growth years, 30 in the third, and 14 in the fourth growth year. The species counts underwent a Hellinger transformation before analysis [73]. The Holm correction [74] was employed to counteract the alpha error inflation due to multiple tests during significance calculations. The analyses were performed using the statistical software CANOCO 5.0 [72].

The relationship between the coverage degrees of the structural variables 'shade', 'herb', 'grass', 'litter', and 'open ground', and the proportions of individuals based on the ecological types and habitat preferences of both animal groups in the SRCs were statistically tested using multiple linear regressions. Additionally, the hypothesis that there is a dependency between vegetation structure diversity and species and habitat preference diversity within ground beetle and arachnid cenoses was tested using simple linear regressions for all study plots and years. Here, the Shannon indices of species and habitat preference diversity were regressed on the Shannon index of the vegetation structure diversity. To verify assumptions of linearity, homoscedasticity, autocorrelation, and normal distribution, studentised residuals were computed from the response variables and tested against the standardised predictor variables. Studentised, excluded residuals from the response variables were used to check for outliers. The SRC residuals exhibited lower dispersions compared to the error values of the reference plots. The presence of

homoscedasticity was assessed using the Goldfeld–Quandt test, allowing for a group-specific examination. Regression coefficients were calculated using the ordinary least squares (OLS) method, which minimises the sum of the squared deviations between observed values and predicted values [75].

To account for potential autocorrelations, the regressions were analysed using a heteroscedasticity- and autocorrelation-consistent estimator (HAC) devised by Newey and West [76]. The Newey–West estimator allows for the specification of a time lag within the dataset. The multiple linear regressions were performed with a time lag of 20 to create a shift for each plot within the dataset for four SRC plots, each with five traps in four study years ($n = 80$). For the simple linear regressions, the arithmetic mean of the diversity values across all traps per plot and year was computed from ten study plots ($n = 40$), incorporating a lag of 10. The interpretation of the coefficient of determination followed Cohen et al. [77], where $R^2 = 0.02$ signifies weak, $R^2 = 0.13$ is moderate, and $R^2 = 0.26$ is high variance explanation. The linear regressions were conducted using the statistical software SPSS v.26.0 [78]. The 'HCREG' macro v.2.0 by Hayes and Cai [79] was used to determine robust Newey–West standard errors, and the 'RRegDiagTest' macro by Grüner [80] was employed for the Goldfeld–Quandt test.

## 3. Results

### 3.1. Vegetation Structure

The vegetation structure in the SRCs consisted of the variables 'herb', 'grass', 'litter', and 'open' (Figure 3). Additionally, these plots were predominantly marked by high degrees of shading. Characteristic for the SCRs were statistically significant differences in the cover levels of the variables between the years, while the reference plots showed smaller differences (Tables S7 and S8).

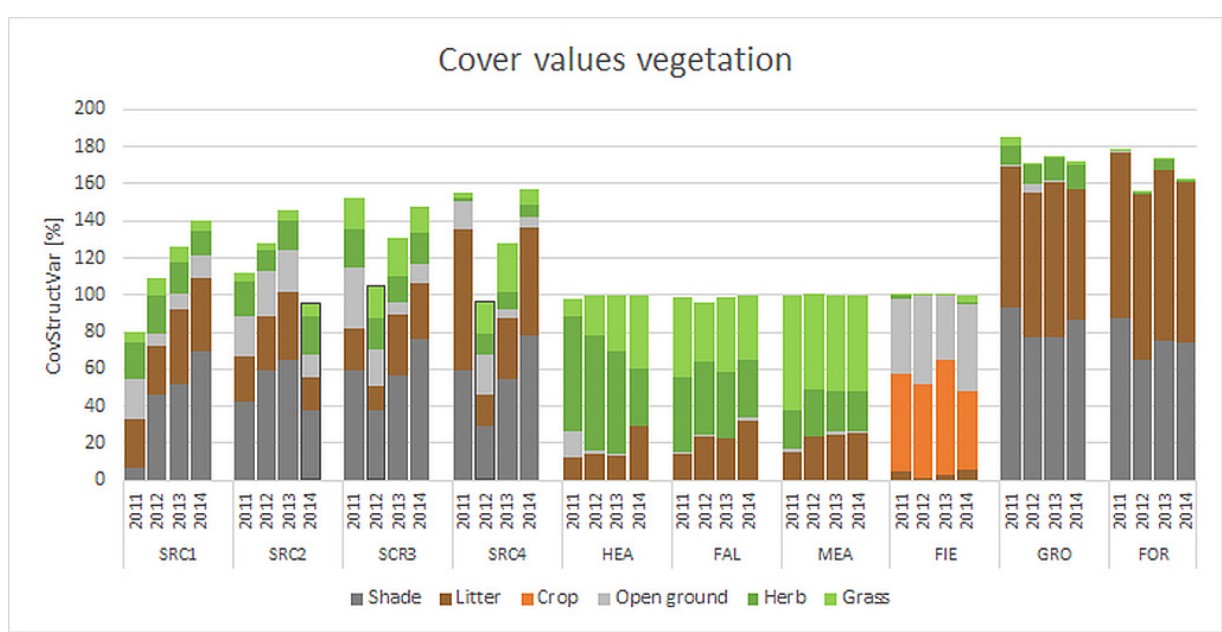

**Figure 3.** Mean percentage cover of selected structural variables (CovStructVar), calculated from 10 survey squares per study plot and year for the period 2011 to 2014. The year after the timber harvest in the SRCs is framed in black. FIE = arable field, FAL = fallow, GRO = grove, SRC1–SRC4 = short-rotation coppices, HEA = headland, FOR = forest, and MEA = meadow. The total degree of cover of the structural variables in the wooded plots SRC1–SRC4, GRO, and FOR may be >100%, as the shade was added to the cover of the other structural variables.

The coverage levels of the variables 'shade' and 'litter' increased during the growth phases within the SRCs, was clearly reduced after timber harvesting, and subsequently

increased again during the subsequent growth phase. During the oldest growth phase, the coverage levels of these two variables in all SRCs were statistically significantly higher compared to the youngest growth phase, while the variable 'open' in SRC1, SRC3, and SRC4 demonstrated a contrary trend (Table S7).

### 3.2. Species and Individual Numbers

A total of 103 ground beetle species in 46,617 individuals were recorded in all study plots between 2011 and 2014. Spiders were found with 167 species and 35,623 individuals, and harvestmen were surveyed with 14 species and 6629 individuals (Tables S1 and S2). The numbers of species and individuals, their mean values, and the associated statistical parameters can be found in Tables S9 and S10. Comparing all plots between 2011 and 2013, the highest number of ground beetle species occurred in one of the SRCs. The fewest ground beetle species and individuals were recorded in the forest in almost all years, with the meadow also characterised by very low species numbers. Except for SRC4 in 2013, the SRCs consistently demonstrated significantly higher average species numbers between 2011 and 2013 compared to the plots FOR, MEA, FAL, and GRO, with minimal differences observed compared to the FIE and HEA plots. In 2014 and throughout the entire study period, the statistically highest average species numbers were found in the headland. In almost all years, there were statistically significant differences in the mean numbers of ground beetle individuals in the SRCs compared to the forest and arable field. On average, throughout the study period, the arable field showed the highest average numbers of ground beetle individuals.

For arachnids, the highest species numbers were recorded in almost all years and throughout the study period in the fallow. The forest exhibited the lowest species numbers between 2011 and 2013, whereas in 2014, this trend was observed in SRC4. Concurrently, the arable field consistently maintained minimal species numbers throughout the entire study period. In almost all years and during the entire study period, the mean numbers of arachnid species in SRC1 and SRC2 showed no statistically significant differences compared to those of the fallow, while SRC3 and SRC4 were consistently lower compared to the fallow. On average, throughout the entire study period, the number of arachnids in the SRCs were statistically significantly lower compared to that in the FIE, HEA, and MEA, and statistically significantly higher compared to the FOR.

### 3.3. Characterisation of Alpha Diversity

The heterogeneity and temporal dynamics of vegetation structure provide the conditions for a diverse range of habitats and a number of potential niches. This can positively impact the colonisation of species with different habitat preferences and the species and habitat preference diversity in both animal groups. The vegetation structure diversity (Shan Expo) per plot and year and the number of ground beetle habitat preferences are graphically represented in Figures 4 and 5.

Three of the SRCs consistently exhibited statistically significantly higher structural diversity compared to the reference plots across all individual years (Figure 4, Table S11). Also, although predominantly not statistically significant, the numbers of ground beetle species with different habitat preferences were generally higher in these SRCs and in the HEA across individual years than in the reference plots (Figure 5, Table S14). Similar high medians were only identified in the fallow in two years and in the arable field in one year. The HEA directly bordered the SRC1 and, like the SRCs in the individual study years, was characterised by distinct changes in the vegetation structure.

For an initial assessment of community structure, the indices log series $\alpha$, Shannon, Shannon exponential, reciprocal Simpson, and evenness were calculated for both species and habitat preference diversity. The statistics for individual years and the entire study duration can be found in Table S12 to Table S15.

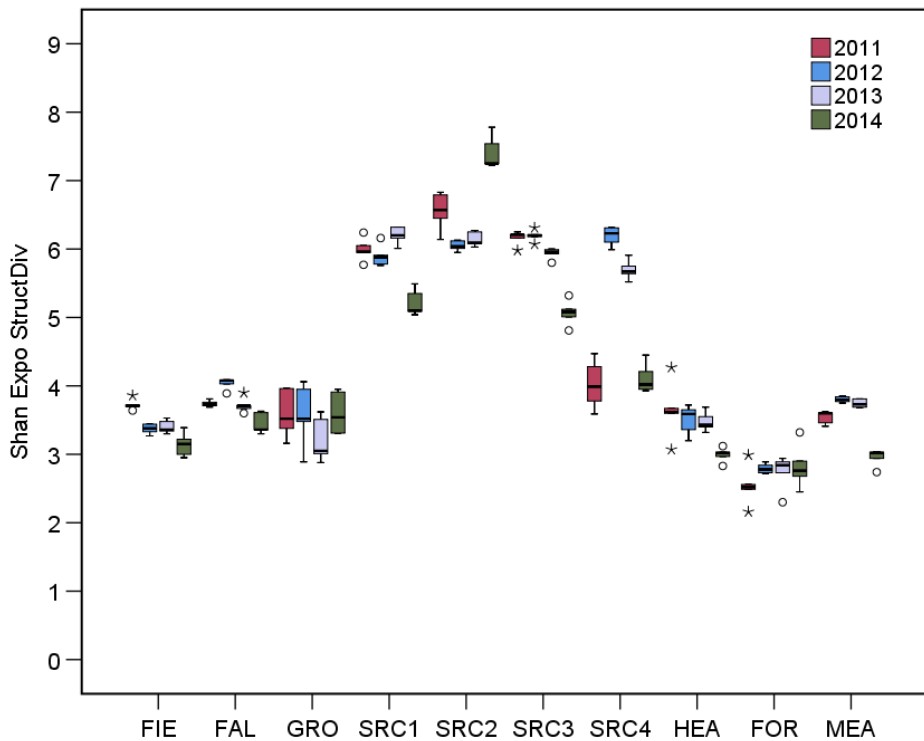

**Figure 4.** Shannon exponential index for vegetation structure diversity (StructDiv) in the individual study years and plots. FIE = arable field, FAL = fallow, HEA = headland, MEA = meadow, GRO = grove, SRC1–SRC4 = short-rotation coppices, and FOR = forest. Circles indicate outliers between 1.5 and 3.0 interquartile range (IQR); asterisks are >3.0 IQR.

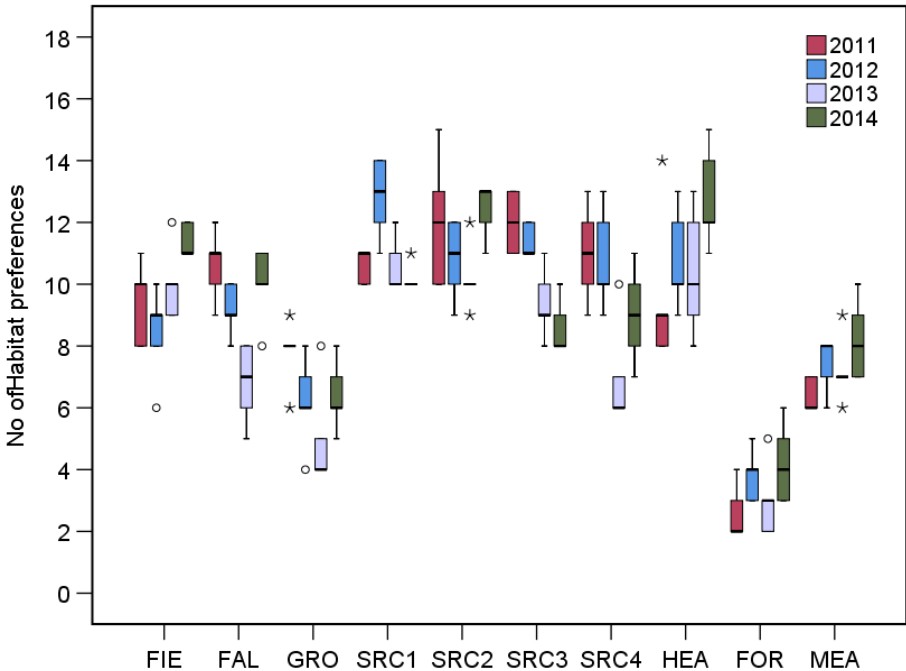

**Figure 5.** Number of ground beetle species with different habitat preferences in the individual study years and plots. FIE = arable field, FAL = fallow, HEA = headland, MEA = meadow, GRO = grove, SRC1–SRC4 = short-rotation coppices, FOR = forest. Circles indicate outliers between 1.5 and 3.0 interquartile range (IQR); asterisks are >3.0 IQR.

Over the entire study period, the ground beetle communities in the SRCs showed significantly higher mean Shannon indices for species diversity than the reference plots, with the exception of SRC4 (Figure 6a). Similarly, the log series $\alpha$ and the reciprocal Simpson index showed consistently higher values in all SRCs compared to the reference plots. In general, the diversity indices, except for one exception, were at their lowest in the meadow across all individual years (Table S12).

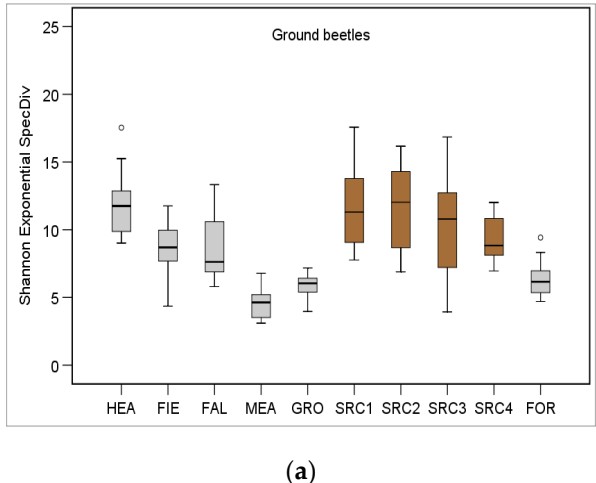
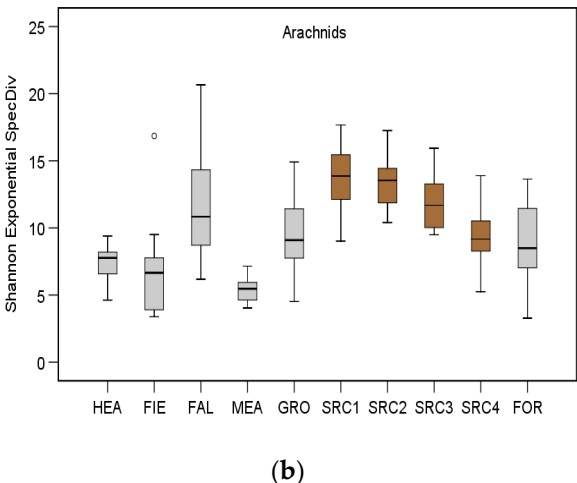

(**a**)  (**b**)

**Figure 6.** Shannon exponential indices of species diversity (SpecDiv) for ground beetles (**a**) and arachnids (**b**). Median for the whole study period 2011–2014. SRC1–SRC4 = short-rotation coppices (brown), HEA = headland, FIE = arable field, FAL = fallow, MEA = meadow, GRO = grove, and FOR = forest (grey). Circles indicate outliers between 1.5 and 3.0 interquartile range (IQR).

The arachnid cenoses showed very heterogeneous species diversity in the different years. However, across all years, the diversity indices attained high to very high values in the SRCs. When comparing the SRCs among themselves, the majority of the cenoses showed statistically significant differences for all diversity measures, on average, over the entire study period (Table S13). Throughout the study period, the Shannon indices for the species diversity in SRCs, except for SRC4, were statistically significantly higher compared to the reference plots (refer to Figure 6b). This trend was consistent for the evenness (along with the forest), the log series $\alpha$, and the reciprocal Simpson index. The lowest diversity values were predominantly found in the forest and arable field.

For the Shannon exponential index of the habitat preference diversity, the ground beetle cenoses showed the highest values in one of the SRCs in the period from 2011 to 2013 (Figure 7a). Similar trends were observed for the log series $\alpha$ and the reciprocal Simpson index, which were mostly statistically significantly higher than those in the reference plots (Table S14). Additionally, high values were also recorded in the fallow for all indices. When comparing the SRCs among themselves, there were no statistically significant differences observed in the habitat preference diversity indices over the entire study period. The statistically significantly lowest diversity values were found in the grove and the forest.

The arachnid cenoses of the SRCs and the fallow showed statistically significantly higher values in the Shannon exponential index of the habitat preference diversity, on average, from 2011 to 2014 compared to those of the other reference plots (Figure 7b, Table S15). The same applied to the reciprocal Simpson index, as well as, in conjunction with the HEA, to the evenness and log series $\alpha$. In terms of diversity measures, there were no statistically significant differences among SRC1, SRC3, and SRC4 throughout the entire study period. In almost all cases, statistically significantly higher values were recorded in SRC2 for the number of habitat preference groups, Shannon and Simpson indices, and evenness. The lowest values for the number of habitat preference groups were mostly found in the forest and grove, and, likewise, the habitat preference indices were predominantly minimal in these plots.

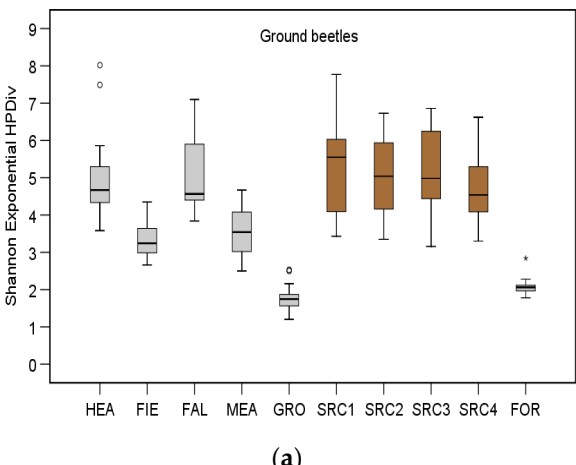
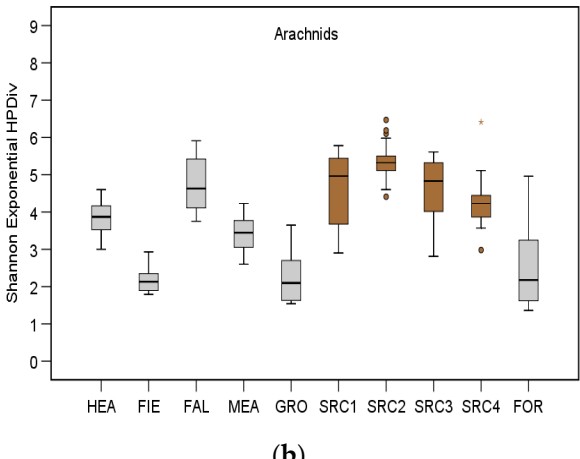

**Figure 7.** Shannon exponential indices of habitat preference diversity (HPDiv) for ground beetles (**a**) and arachnids (**b**). Median for the study period 2011–2014. SRC1–SRC4 = short-rotation coppices (brown), HEA = headland, FIE = arable field, FAL = fallow, MEA = meadow, GRO = grove, and FAL = forest (grey). Circles indicate outliers between 1.5 and 3.0 interquartile range (IQR); asterisks are >3.0 IQR.

### 3.4. Impacts of Vegetation Structure on Communities

The impact of changes in vegetation structure during individual growth years and after timber harvesting on the quantitative and qualitative composition of ground beetle and arachnid cenoses was differentiated by each growth year using redundancy analyses (RDAs) for all study plots. These were tested and are graphically depicted in Figures 8a–d and 9a–d. For clarity, individual species positions were omitted from the graphs to highlight differences in the weighted compositions of the ground beetle and arachnid communities along environmental gradients.

A pseudo-canonical correlation of more than 90% existed between the species data and structural variables on the first two axes in both animal groups (Tables S16 and S17). These two axes explained approximately half of the total variance in the ground beetle species data and at least 45% in the arachnid species data. The variable 'shade', which was positively associated with the first axis, consistently demonstrated high statistical significance in explaining the major variance across all growth years for both animal groups (Tables S18 and S19). Similarly, the variable 'litter' exhibited a positive correlation with the first axis and made a statistically significant contribution to explaining the variance. Additionally, the variable 'grass' substantially contributed to explaining the total variance in the species data for both animal groups on the second axis (Tables S16 and S17). Regression coefficients for the relationships between the species data and structural variables on the canonical axes for each growth year can be found in Tables S20 and S21 for both animal groups.

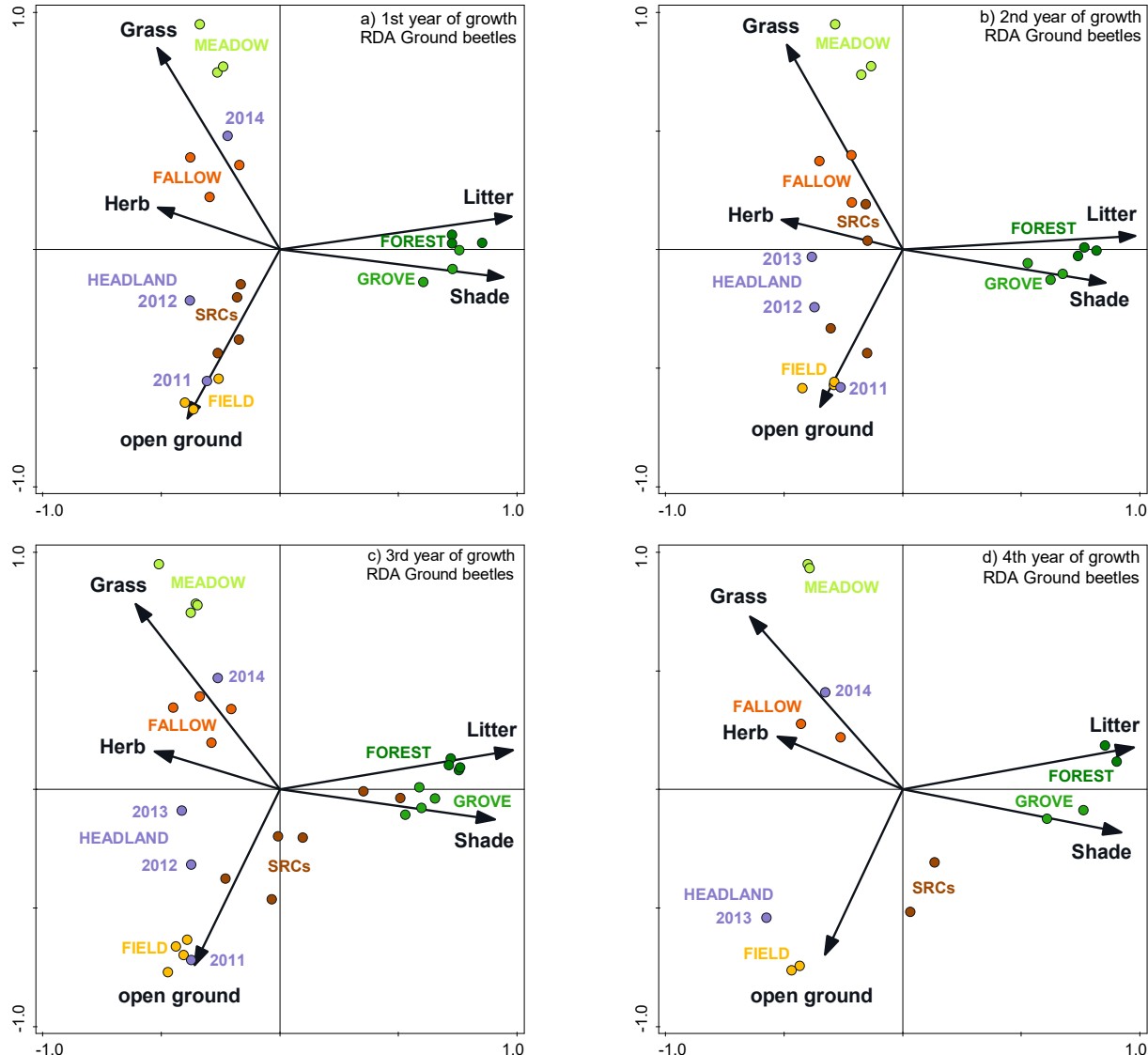

**Figure 8.** Ordination diagrams (1st and 2nd ordination axis) of ground beetles based on redundancy analyses (RDAs). Shown are the communities represented by plot symbols (annual sums of individuals from five pitfall traps per plot) in the plots SRC1–SRC4 per year of growth (**a**–**d**) as well as the reference biotopes examined in the same period in relation to five structural variables. The plot points were colour-coded and labelled with the corresponding colour.

Although the plot points of the SRC approached those of individual reference plots in different growth years, the species communities exhibited distinct differences in the composition of the variables relevant to them compared to those of the reference plots.

In the first two growth years, negative loadings were present on the first axis for the species communities of the SRCs in both animal groups (Tables S18 and S19). A relatively high proportion of individuals of open-field species characterised the communities of the SRCs during these years. In the third growth year, positive loadings on the third axis, which were positively correlated with the variable 'herb', were observed for the species communities of almost all SRCs, as well as those of the fallow and the headland. However, there were only limited similarities among the communities of these plots. In the third and fourth growth years, the communities showed higher proportions of forest species and displayed positive loadings on the first axis, positively correlated with the variables 'shade' and 'litter'. Similar to the SRCs, a change in the ground beetle and arachnid communities

was also evident in the headland over time, which was accompanied by a change in the species spectrum characteristic of perennial wildflower seedings.

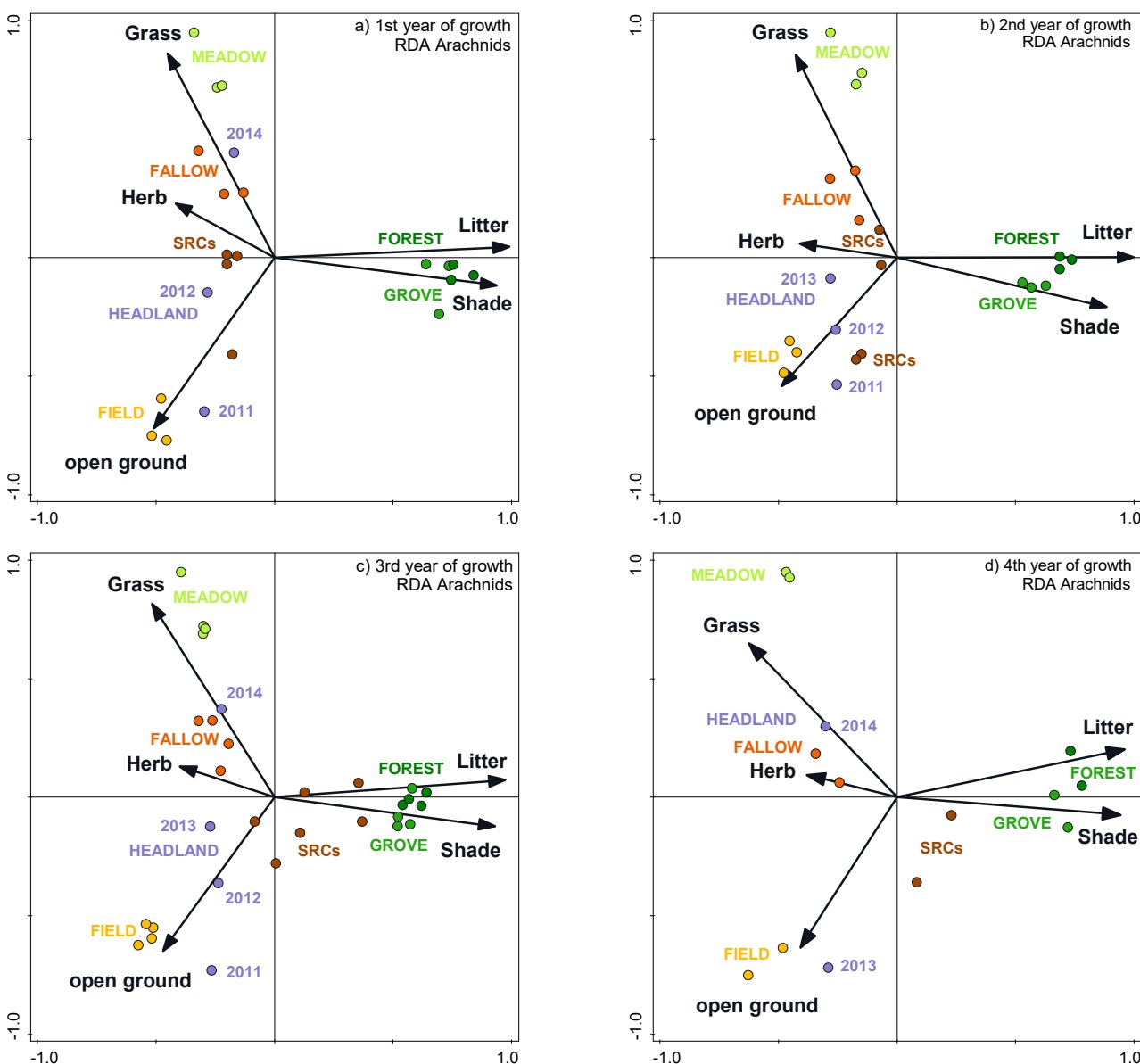

**Figure 9.** Ordination diagrams (1st and 2nd ordination axis) of arachnids based on redundancy analyses (RDAs). Shown are the communities represented by plot symbols (annual sums of individuals from five pitfall traps per plot) in the plots SRC1–SRC4 per year of growth (**a**–**d**) as well as the reference biotopes examined in the same period in relation to five structural variables. The plot points were colour-coded and labelled with the corresponding colour.

### 3.5. Cenoses of the SRCs in the Individual Study Years

During the study period, there were major changes in the degree of cover of the structural variables for the SRCs, which were accompanied by changes in the composition of the species communities and their habitat preferences in both animal groups. Figure 10a,d exemplifies the development of the variable 'shade' for a plot with and without timber harvesting during the study period. During the growth phases, the shading levels and litter content in the SRCs statistically significantly increased each year (Table S7). Additionally, the forest species in both animal groups exhibited higher proportions of individuals in each growth year compared to the preceding year (Figure 10b,c,e,f and Tables S24 and S25).

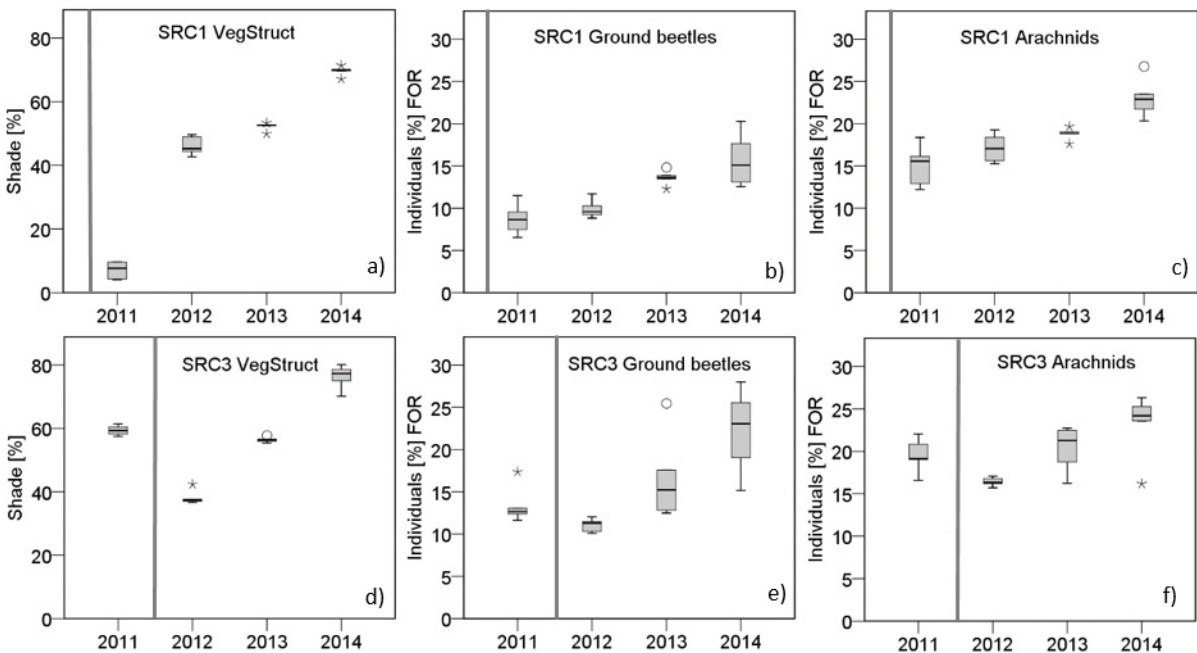

**Figure 10.** Shade cover in the plots SRC1 (**a**) and SRC3 (**d**) and percentage of individuals of general forest species (FOR) of the ground beetle (**b,e**) and arachnid (**c,f**) communities for the study years 2011–2014. Circles indicate outliers between 1.5 and 3.0 interquartile range (IQR); asterisks >3 IQR. Statistically significant differences between the years can be seen in Tables S22 and S23. The vertical bars indicate timber harvest.

In the year following timber harvesting, the mean coverage levels of shading and litter statistically significantly decreased in all SRCs (see Figure 10d and Table S7). This was associated with lower proportions of forest species in both animal groups compared to the previous year (Figure 10e,f), while an opposing trend was observed for open-field species (Table S24 and S25). In the second year after timber harvesting, the SRCs exhibited high coverage levels again for the structural variables 'shade' and 'litter'. Forest species of ground beetles dominated the cenoses in all SRCs in the oldest growth year, while this was characteristic of arachnids in almost all years.

In contrast to the youngest growth year, the forest species of both animal groups in all SRCs exhibited statistically significantly higher mean individual proportions in the oldest growth year, while for the arable and grassland species (Figure 11a–d), the opposite was observed in three SRCs (Tables S22 and S23).

The changes in habitat structure also affected the qualitative composition of the forest species in all SRCs. From the first to the oldest growth year, higher proportions of individuals were observed each year for stenotopic forest species of arachnids compared to the previous year, while lower proportions were found for eurytopic forest species (Figure 12a–d). This was also true for three SRCs in the case of ground beetles, and in the oldest growth year, the mean proportions of individuals of stenotopic forest species for both animal groups were statistically significantly higher than in the youngest growth year (Tables S28 and S29).

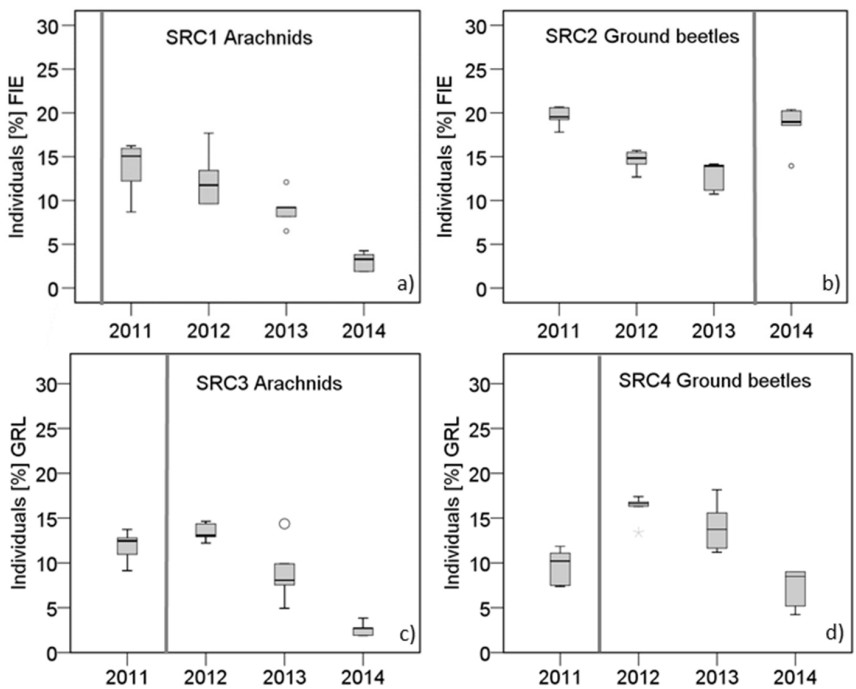

**Figure 11.** Percentage of individuals of arable (FIE) (**a**,**b**) and grassland species (GRL) (**c**,**d**) of the arachnid and ground beetle communities in the plots SRC1–SRC4 for the study years 2011–2014. Circles indicate outliers between 1.5 and 3.0 interquartile range (IQR); asterisks >3 IQR. Statistically significant differences between the years can be seen in Tables S22 and S23. The vertical bars indicate timber harvest.

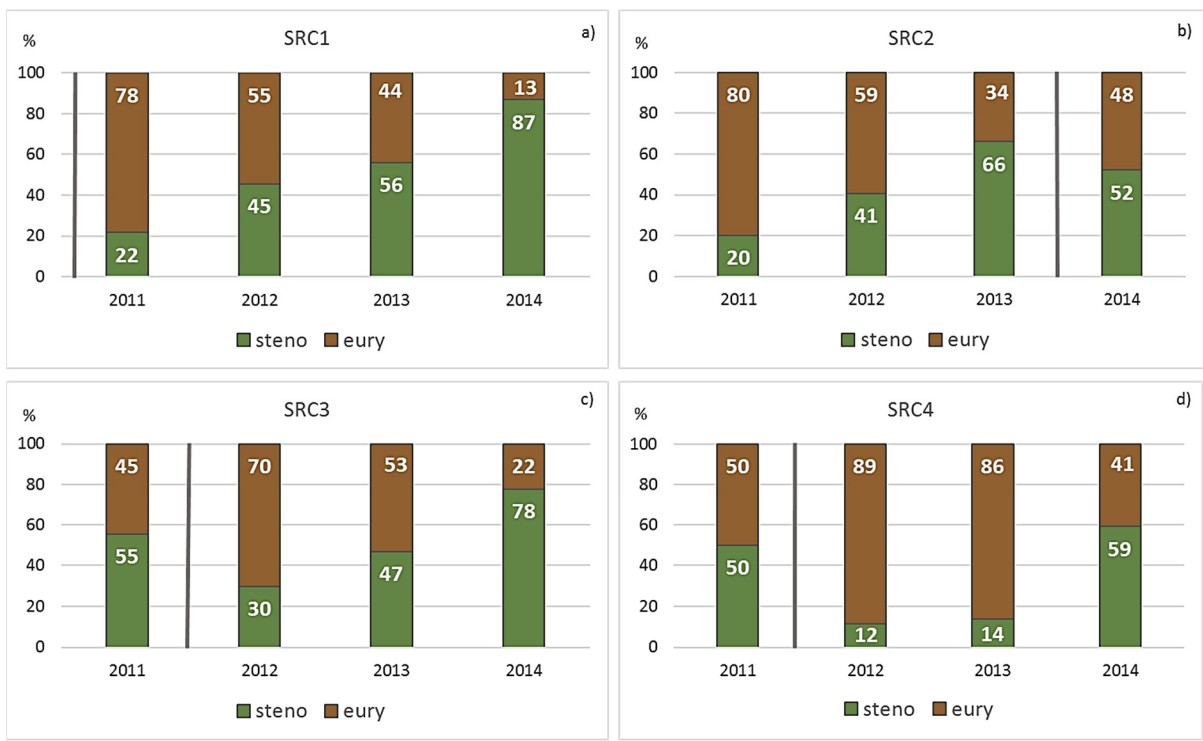

**Figure 12.** Percentage of individuals of stenotopic (green) and eurytopic (brown) forest species for the arachnids in the plots (**a**) SRC1, (**b**) SRC2, (**c**) SRC3, and (**d**) SRC4. The individuals of the stenotopic (steno) forest species include moist forest (MFO), low mountain forest (LMF), wet forest (WFO), and acidophilous forest (AFO) species, and those of the eurytopic (eury) forest species were preliminary forest species (PFO) and forest species not bound to a specific forest type (FOR) and were totalled for five traps per study year. The time of timber harvesting is indicated by a vertical black line.

### 3.6. Cenoses of the SRCs throughout the Study Period

The determination coefficients ($R^2$) confirmed a high variance explanation by the structural variables in the ecological groups for both animal groups in almost all analyses. The proportions of grassland species in both animal groups and those of xerophilic open-land species among the arachnids were statistically highly negatively correlated with the variable 'shade' (Figure 13a,b and Table S31 to Table S33).

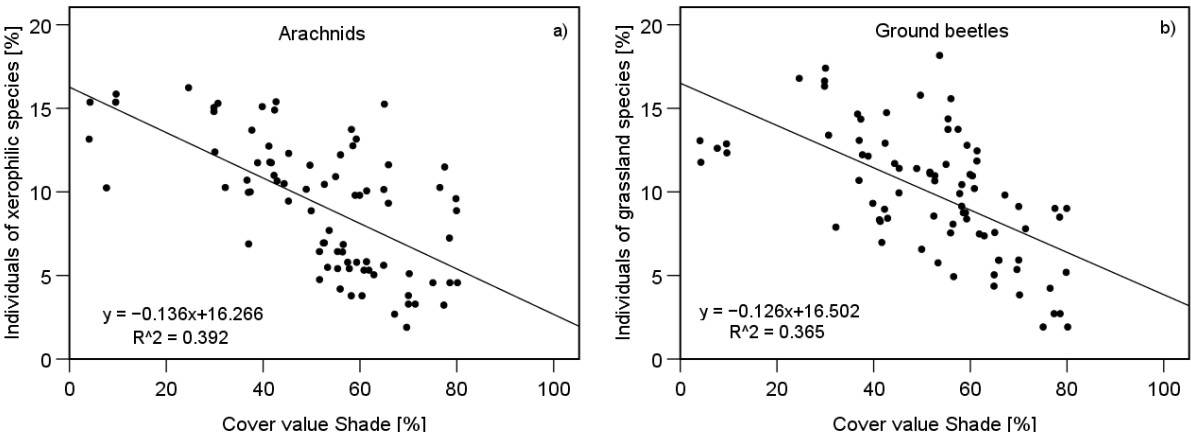

**Figure 13.** Relation between the degree of shading and the percentage of individuals of xerophilic (x) arachnid species (**a**) and grassland species (GRL) of ground beetles (**b**). Linear simple regression for the plots SRC1–SRC4 in the study years 2011–2014 on the basis of sums of individuals from five traps per plot and year (*n* = 80).

In contrast, the proportions of the ecological type of weakly hygrophilic forest species '(h)f' for both animal groups (Figure 14a for arachnids) and those of eurytopic forest species of ground beetles showed statistically highly significant positive relations with the variable 'shade' (Tables S30 and S32).

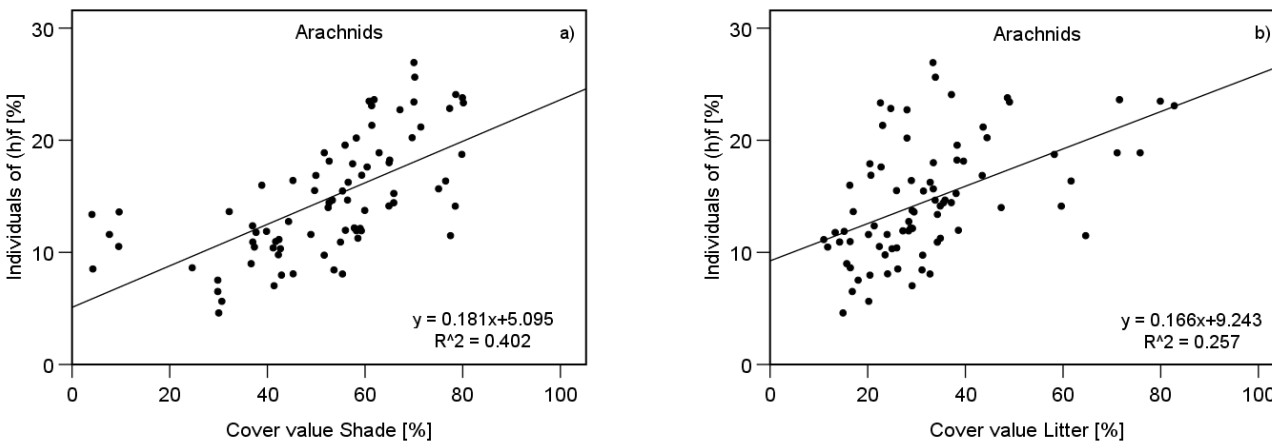

**Figure 14.** Relation between the percentage of individuals of arachnid species with the ecological type 'slightly hygrophilic forests' ((h)f) and the degree of shade cover (**a**) and the degree of litter cover (**b**). Linear simple regression for the plots SRC1–SRC4 in the study years 2011–2014 on the basis of sums of individuals from five traps per plot and year (*n* = 80).

With respect to the variable 'litter', a statistically significant to highly significant positive correlation was observed for the proportions of weakly xerophilous and weakly hygrophilous forest species among the arachnids (Figure 14b) and those of eurytopic forest species of ground beetles. The proportions of hygrophilic open-land species showed a statistically highly significant negative relationship with litter coverage for both animal

groups. Additionally, the proportions of arable species and those of xerophilic and eurytopic open-land species among the arachnids also showed a statistically significant to highly significant negative relationship with litter coverage (Tables S30, S32, and S33).

### 3.7. Structural and Ecological Diversity

For both animal groups, there existed a statistically highly significant positive linear relationship between vegetation structure diversity and species and habitat preference diversity (Table 1 and Figure 15a–d). Structural diversity contributed to approximately 30% for ground beetles and around 26% for arachnids in explaining the species diversity. An increase in the structural diversity by one unit was associated with an increase in the species diversity by more than half a unit in both animal groups.

**Table 1.** Statistical parameters of the relationships between the species diversity (SpecDiv) and habitat preference diversity (HabDiv) (Shannon index) of the ground beetle and arachnid cenoses and the vegetation structure diversity in the plots of the years 2011–2014 on the basis of a linear regression. Shan = Shannon index, $R^2$ = coefficient of determination, B = regression coefficient, F = F-value, df = degrees of freedom, $p$ (model) = significance of the model, SE = standard error, t = $t$ value, $p$ (coeff.) = significance of the coefficients, and HAC = statistical parameters calculated using robust Newey–West estimators.

| | Ground Beetles | | Arachnids | |
|---|---|---|---|---|
| | **Shan SpecDiv** | **Shan HabDiv** | **Shan SpecDiv** | **Shan HabDiv** |
| | Model overview | | | |
| $R^2$ | 0.297 | 0.380 | 0.257 | 0.445 |
| B | 0.662 | 0.834 | 0.622 | 0.814 |
| F | 16.081 | 23.311 | 13.123 | 30.528 |
| $F_{HAC}$ | 22.676 | 47.613 | 109.518 | 67.629 |
| df | 1. 38 | 1. 38 | 1. 38 | 1. 38 |
| $p$ [Model} | <0.001 | <0.001 | 0.001 | <0.001 |
| | Coefficients | | | |
| SE (B) | 0.165 | 0.173 | 0.172 | 0.147 |
| t | 4.010 | 4.828 | 3.623 | 5.525 |
| $t_{HAC}$ | 4.762 | 6.900 | 10.465 | 8.224 |
| $p_{HAC}$ [Coeff.] | <0.001 | <0.001 | 0.001 | <0.001 |

Similarly, there was a strong positive linear relationship between habitat preference diversity and vegetation structure diversity for both animal groups (Table 1 and Figure 15c,d). The vegetation structure diversity significantly explained the total variance of the habitat preference diversity ($p < 0.001$), contributing about 40% for both animal groups. The habitat preference diversity increased by a factor of 0.8 in both groups with an increase of one unit in structural diversity. Thus, structural diversity held greater importance for habitat preference diversity compared to species diversity in both animal groups.

A high heterogeneity in the vegetation structural characteristics was linearly correlated with high species diversity in the ground beetle and arachnid cenoses in the SRCs. Furthermore, higher structural diversity in vegetation encouraged the suitability of SRCs for a large number of species with different habitat preferences.

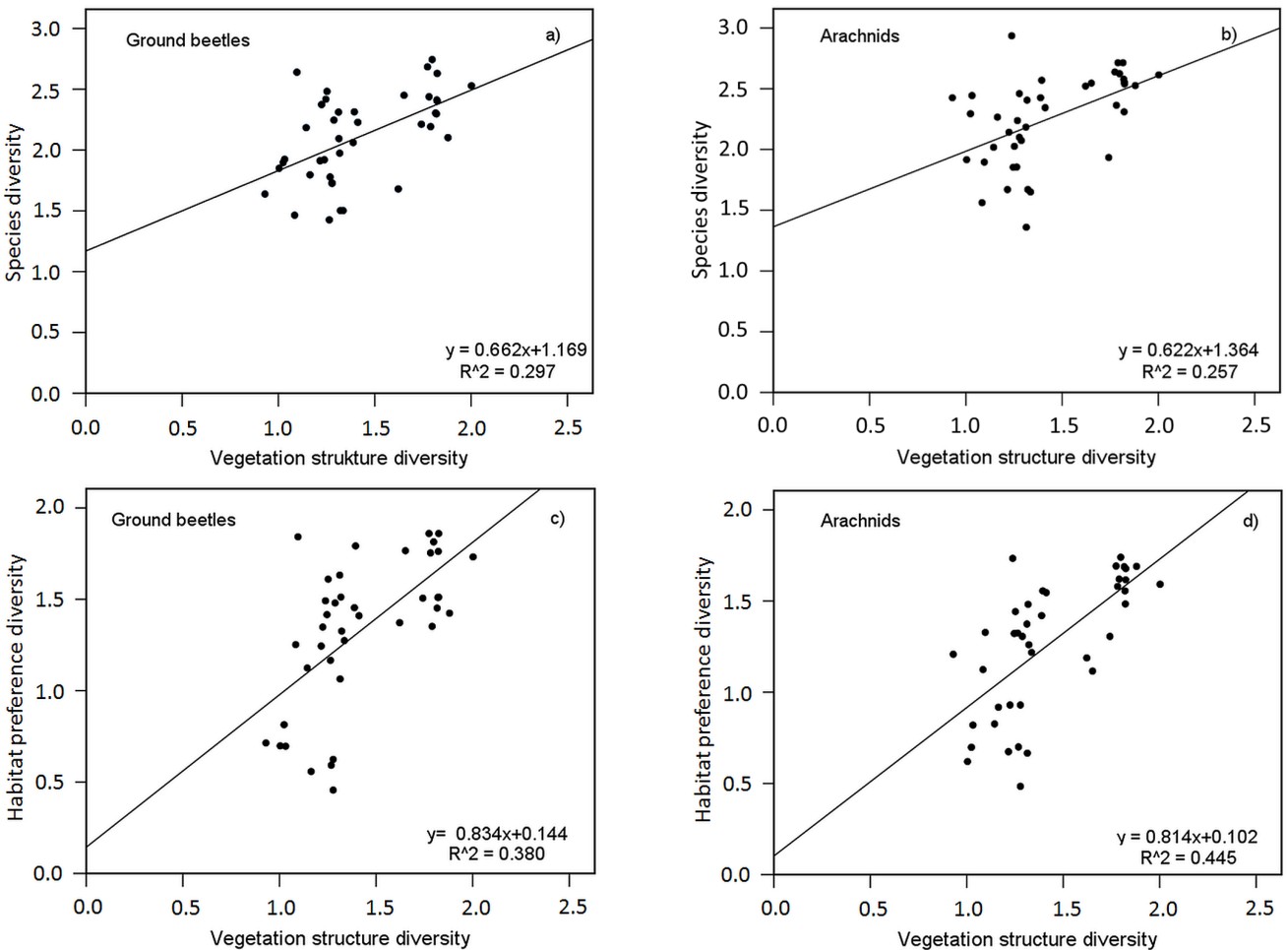

**Figure 15.** Relation between the Shannon indices of vegetation structure diversity and the Shannon indices of species and habitat preference diversity of ground beetles (**a**,**c**) and arachnids (**b**,**d**) in SRC1 to SRC4 for the study years 2011–2014. Linear regression based on the diversity values per plot and year (*n* = 40).

## 4. Discussion

### 4.1. Composition of Communities

Our studies revealed comparatively high species diversity for the ground beetle and arachnid cenoses in the SRCs compared to those in the reference plots. This finding is supported by several studies [81–84] for ground beetles, as well as by Blick et al. [85] for arachnids. Assessing SRCs for their species diversity and their role in enhancing diversity in the surrounding agricultural landscape can be significantly influenced by the selection of reference biotopes, as emphasised by Glemnitz et al. [31]. Allegro and Sciaky [86], for instance, found a higher number of ground beetle species in SRCs in comparison to forests, while other authors reported lower species counts compared to fields [87–89]. Müller-Kroehling et al. [32] also highlighted that comparing species counts disregards qualitative aspects such as rare, endangered, or specialised species, and, hence, only holds validity within comparable land use forms. In agreement with the findings of Schulz [90], Šťastná [91], and Ulrich et al. [89], there are great commonalities in the species composition of ground beetle and arachnid cenoses between SRCs and reference biotopes. SRCs do not provide a suitable habitat for rare or endangered ground beetle or arachnid species, as confirmed by [89,91–94]. Müller-Kroehling et al. [32] only identified endangered pioneer species in SRCs during the establishment phase. For rare species, the authors highlighted the importance of edge areas and disturbance spots, which are more characteristic of small,

non-commercially exploited SRCs [29]. Similarly, Kriegel et al. [95] identified the highest number of rare species in stand gaps.

### 4.2. Cenotic Changes during SRC Woody Growth

Both in terms of age since planting and in relation to growth years, SRCs undergo specific developments [32]. Characteristic of SRCs are the changes in the vegetation structure's variable coverage associated with each growth year. Over the course of growth years, the cenoses of both animal groups in the SRCs consistently show a shift in composition. For the forest species of both animal groups, there are higher proportions each year during the growth phase compared to the previous year, and they dominate the cenoses in all SRCs in the oldest growth year. A reverse trend is evident for the proportions of open-land species. Schulz et al. [29], Allegro and Sciaky [86], and Nerlich et al. [96] also observed an increase in the number of forest species and a decrease in open-land species of ground beetles in SRCs. Schardt et al. [97] confirmed the same changes for arachnid cenoses, and Glemnitz et al. [31] noted an increase in the number of individuals of forest species in strip-like SRCs. The changes in the quantitative and qualitative species compositions during the study period indicate the development of unique cenoses in SRCs, occurring rapidly for both animal groups. A development of specific cenoses in SRCs was also confirmed by Boháč et al. [81], Havlíčková and Rudišová [82], Al Hussein et al. [98], Liesebach and Mecke [99], and Verheyen et al. [84] for ground beetles, and by Blick and Burger [93], Schardt et al. [97], and Burger [100] for arachnids.

Within four years, SRCs undergo a transition from open to wooded habitats [28]. Three years after planting, the cenoses of ground beetles and arachnids in the SRCs already show very high proportions of forest species. Colonisation by forest species in the densely wooded study region could be supported by the proximity to forest biotopes. The distance between SRC4 and the nearest forest is about 400 m. According to Schulz et al. [29], the distance to woodland habitats can significantly favour the colonisation of SRCs by forest species. Boháč et al. [81], Havlíčková and Rudišová [82], and Nerlich et al. [96] also identified typical forest species among ground beetles within the first four growth years. In contrast, Sachs et al. [88], Schardt et al. [97], Burger [100], Lamersdorf et al. [101], and Schulz et al. [29] observed forest species among ground beetles and arachnids in SRCs only after eight years. Even after 10 years in completely isolated SRCs, Allegro and Sciaky [86] found few forest species among ground beetles in the Italian Po Valley.

In a nine-year study, Gruttke and Willeke [102] used ground beetles, arachnids, and woodlice (Isopoda) to show that the colonisation capacity of newly planted woody lands by forest species is considerably influenced by the proximity to larger forests and that the plantations should, therefore, not be planted arbitrarily in cleared, intensively farmed agricultural landscapes. This is corroborated by Müller-Kroehling et al. [32] for strict forest species. The authors found a statistically significant correlation between the number of strict forest species and the distance to the nearest forest. Besides proximity to woodland habitats, the macroclimate can also influence the colonisation of SRCs by forest species, as shown by Brauner and Schulz [94], Al Hussein et al. [98], and Weger et al. [92] for dry–warm regions. The regional species pool's composition [90] and reduced management intensity compared to one-year crops [103] are additional factors influencing the colonisation of SRCs by invertebrates.

### 4.3. Vegetation Structure's Impact on Ecological Species Traits within Communities

The authors of various studies have emphasised that the habitat selection of ground beetles and spiders is significantly influenced by vegetation structure, e.g., [104–110]. Throughout the study period, three SRCs consistently exhibited statistically significantly higher structural diversity compared to the reference biotopes as well as in all individual years. The high number of distinct habitat preference groups in both animal groups indicated that this structural heterogeneity within the SRCs can positively impact the colonisation of species with different habitat preferences.

As the structural composition of a habitat is largely determined by vegetation, it directly and indirectly influences the habitat characteristics [111]. Verheyen et al. [84] established a positive correlation between structural heterogeneity and the diversity of ground beetle and spider species with differing ecological traits within SRCs. Our analyses confirmed these findings, demonstrating a statistically significant relationship between structural habitat characteristics, ecological traits, and the specific habitat requirements of the species. In particular, the degree of shading and the proportion of litter statistically significantly increased in the SRCs during the growth phase. The mean proportions of forest species exhibited statistically significantly positive correlations with these variables in both animal groups. Conversely, negative correlations ($p < 0.001$) existed in both animal groups for the mean proportions of arable and grassland species with shading. Several studies [112–118] also emphasised that shading strongly influences species composition. The positive impact of litter is particularly highlighted by [105,119–123], as the number of microhabitats increases, and species diversity significantly rises with the greater thickness and complexity of the litter [124].

Insights into the relationship between various hunting strategies and the spatial structure of the habitat are provided by the studies of Brose [111], Brunk [112], Bonn and Kleinwächter [125], and Diehl et al. [126]. For ground beetles, the potential prey availability increases with a greater complexity of vegetation structure [127]. Conversely, Kalinkat et al. [128] demonstrated through experiments with centipedes (Chilopoda) that the capturing success decreases with an increased complexity of vegetation because the likelihood of predator and prey encounters diminishes. However, for ground beetles and arachnids, SRCs may offer a greater food supply compared to arable land, as suggested by the results of Britt et al. [87], Burmeister and Walter [129], Liesebach et al. [130], and Rowe et al. [131].

### 4.4. Effects of Timber Harvesting

Following several years of growth of SRCs, timber harvesting leads to sudden and profound changes in habitat. The microclimatic conditions (solar radiation, moisture, and temperature) and the structural features of the habitat, affecting the living conditions of the resident fauna, undergo significant alterations. With the timber harvest, the trend of escalating proportions of forest species within the SRCs concludes. Both animal groups partly exhibited forest species with statistically significantly lower proportions in the subsequent year after a harvest compared to the previous year, while open-land species showed statistically significantly higher proportions. These changes were accompanied by a statistically significant decrease in the degree of coverage of the variables of 'shade' and 'litter' compared to the previous year.

Despite the decline in the proportions of forest species in both animal groups in the year following timber harvesting compared to the previous year, ground beetles exhibited proportions of between 10% and 20%, and arachnids showed proportions of between 20% and 30% in these SRCs during the subsequent year. These relatively high proportions suggest that after a timber harvest, a complete recolonisation by forest species does not occur, indicating that at least a portion of these animals already or still reside within the SRCs (the 'sawtooth hypothesis' by Platen et al. [132] in Veste and Böhm [27]). This suggests the potential of rotationally harvested SRCs as a connecting element between wooded habitats in agricultural landscapes within reasonable distances.

In the second year after timber harvesting, the variables 'shade' and 'litter' once again reached high cover levels, and the trend of higher proportions of forest species compared to the previous year, along with lower proportions of open-land species, continued. The mean proportions of arable and grassland species in both animal groups were statistically significantly lower in the oldest growth year than in the youngest. In contrast, the general forest species, particularly the stenotopic forest species, showed statistically significantly higher proportions in the oldest growth year compared to the youngest. This indicates that timber harvesting in a rotational way effectively supports the colonisation of SRCs by forest species of ground beetles and arachnids.

*4.5. Structural Diversity's Impact on Ecological Variety*

Our studies showed that the Shannon index of the species and habitat preference diversity was positively correlated ($p \leq 0.001$) with the vegetation structure diversity for the cenoses of both animal groups in the SRCs. The species and habitat preference diversity within the SRCs was statistically significantly higher in both animal groups compared to the reference plots.

According to Loreau et al. [133], diverse species' reactions to environmental variations have a stabilising effect on ecosystem functioning. Higher species diversity increases the likelihood of the presence of species with diverse ecological requirements and traits, where the variability in species' reactions to environmental fluctuations directly measures functional diversity [134]. Species capable of utilising previously untapped resources at present due to their ecological requirements and traits enhance resource efficiency and functionally contribute to the ecosystem. The high habitat preference diversity within the SRCs reflected a considerable number of species with diverse ecological tolerances for both animal groups. In highly dynamic habitats like SRCs, the presence of species with various habitat preferences can be advantageous, potentially allowing responses to abrupt and profound habitat changes within shorter periods, contrary to resettlement (Insurance Hypothesis [135]). Moreover, typical species characteristic of agricultural landscapes and exhibiting a broad ecological range settle within SRCs. This also bears functional significance, as a wider range of species reactions results in reduced variability or a greater buffering effect within the ecosystem [136]. Both aspects contribute to efficient resource utilisation.

Furthermore, the high habitat preference diversity also signifies the ecological equivalence of species with similar ecological requirements. A relatively high number of different forest species suggests a more or less pronounced niche overlap within this habitat preference group, reflected in the partial different dominance of species within this habitat preference group in individual years. Species with similar ecological traits can enhance the productivity of ecosystems through niche complementarity, thereby reducing competition [133]. For instance, based on their body size, they may possess different feeding habits or employ various hunting strategies. Additionally, due to their activity, different predatory species encounter distinct prey at specific times of the day and seasons. Thus, species of the same functional type with different ecological requirements and tolerances contribute to the ecological equivalence of essential ecosystem functions [136,137] and signify 'insurance of functionality in the future' [138]. The comparatively high proportions of open-land species in the year following the timber harvest compared to the previous year, alongside the renewed high proportions of forest species in the subsequent year, underscore the resilience of the cenoses and confirm the fundamental suitability of SRCs as habitats for resilient ground beetle and arachnid communities.

Loreau et al. [133] demonstrated that high complementarity and low selection effects can improve the stability of ecosystem productivity. The authors emphasised that by diversifying agricultural systems over time (e.g., crop rotation) and space (e.g., catch crop), insurance effects can occur, which can be considered a partial substitute for crop insurance. Heterogeneous landscapes, where ecosystems are interconnected through the movements of organisms, maintain a high alpha diversity through spatial complementarity among species and contribute, at the regional level, to reducing the temporal variability of ecosystem properties (Spatial Insurance Theory [139]). Differentiated, long-term studies on the impact of vegetation structure on the composition of ground beetle and arachnid cenoses in commercially managed SRCs have not yet been conducted. Our findings suggest that SRCs are characterised by high habitat heterogeneity and can contribute to the diversification of agricultural landscapes. SRCs can support natural pest regulation via quantitatively important regulators such as ground beetles and arachnids. SRCs can provide a positive contribution to maintaining the diversity of ground beetle and arachnid communities in agricultural landscapes, particularly in rotational harvesting.

## 5. Conclusions

SRCs are characterised spatially and temporally by a high habitat heterogeneity. For both animal groups, the composition of communities in the SRCs changed during the growth phase and after timber harvesting. Correlations between ecological species traits and the cover levels of selected structural variables of vegetation confirmed that the habitat selection of ground beetles and arachnids in SRCs is highly influenced by the vegetation structure. The communities of both animal groups in the SRCs generally exhibited higher species and habitat preference diversities than the reference biotopes. These were positively correlated with the structural diversity of the SRCs, which was significantly higher compared to the reference biotopes. This reveals that the habitat heterogeneity of SRCs can positively affect species diversity and the colonisation of species with different habitat preferences. We conclude that SRCs can contribute to the structural diversification and maintenance of ground beetle and arachnid diversity in agricultural landscapes. In SRCs with timber harvesting, individual proportions between 10% and 20% for the forest species of ground beetles and 20% and 30% for arachnids were recorded the following year. This indicates that complete recolonisation by forest species does not occur after a timber harvest and that rotational timber harvesting effectively supports colonisation by forest species of ground beetles and arachnids. We conclude from this and from the generally high proportions of individuals of forest species during the regrowth years that SRCs are suitable as a connecting element for ground beetles and arachnids between woodland biotopes in agricultural landscapes within reachable distances.

**Supplementary Materials:** The following supporting information can be downloaded at: https://www.mdpi.com/article/10.3390/land13020145/s1, Table S1. List of ground beetle species and individuals (Ind) of the study plots in the period 2011–2014 with indication of ecological type (ET) and habitat preference (HP). Legends: see Tables S3 and S5. Table S2. List of arachnid species and individuals (Ind) of the study plots in the period 2011–2014 with indication of ecological type (ET) and habitat preference (HP). Legends: see Tables S3 and S5. Table S3. List of differentiated ecological types (ETs) after [64] and [65], modified. Table S4. Summed groups of ecological types based on the differentiated ecological types (DiffETs) according to Table S3. Table S5. List of differentiated habitat preferences (DiffHPs) according to [64,65,67], modified. Table S6. Summerised habitat preference groups according to Table S5. Table S7. Mean percentage cover of selected structural variables in the SRC plots, calculated from 10 survey areas of 1 m$^2$ for each survey year. Yog = year of growth, Rot = rotation phase, and SE = standard error. One-factorial ANOVA (Duncan's test, $p \leq 0.05$); degrees of freedom (df): 3.16; F = F-value; a–d = statistically significant differences between plots. Table S8. Mean percentage cover of selected structural variables in the reference plots, calculated from 10 survey areas of 1 m$^2$ for each survey year. Yog = year of growth, Rot = rotation phase, and SE = standard error. One-factorial ANOVA (Duncan's test, $p \leq 0.05$); degrees of freedom (df): 3, 16; F = F-value; a–d = statistically significant differences between plots. SE = standard error, FIE = arable field, FAL = fallow, GRO = grove, HEA = headland, FOR = forest, and MEA = meadow. Table S9. Absolute numbers (a) and mean values (m) of species and individuals of ground beetles in all plots for the individual years 2011 to 2014, as well as for the entire study period. For the SRC plots, the year of growth is given in brackets, cells with maximum values are highlighted in green, and cells with minimum values in yellow. Species = number of species, individuals = number of individuals, SE = standard error. One-factorial ANOVA, based on the sum of individuals from five single traps per study year. Degrees of freedom (df): 9.40; F = F-value; a–f indicate statistically significant differences ($p \leq 0.05$) between the plots. FIE = arable field, FAL = fallow, GRO = grove, SRC1–SRC4 = short-rotation coppice plots, HEA = headland, FOR = forest, and MEA = meadow. Table S10. Absolute numbers (a) and mean values (m) of species and individuals of arachnids in all plots for the individual years 2011 to 2014, as well as for the entire study period. For the SRC plots, the year of growth is given in brackets, cells with maximum values are highlighted in green, and cells with minimum values in yellow. Species = number of species, individuals = number of individuals, and SE = standard error. One-factorial ANOVA, based on the sum of individuals from five single traps per study year. Degrees of freedom (df): 9.40; F = F-value; a–g indicate statistically significant differences ($p \leq 0.05$) between the plots. FIE = arable field, FAL = fallow, GRO = grove, SRC1–SRC4 = short-rotation coppice plots, HEA = headland, FOR = forest, and MEA = meadow. Table S11. Rounded averages of the vegetation

structure diversity indices Log series $\alpha$, Shannon exponential index (Shan-Expo), Shannon index, reciprocal Simpson index, and evenness, calculated from the values of ten single vegetation survey squares per plot and year, and for the entire study period. The year of growth for the SRC plots is given in brackets. One-factorial ANOVA (Duncan's test, $p \leq 0.05$); degrees of freedom (df): 9.40; F = F-value; a–g indicate statistically significant differences between the plots. Cells with maximum values are highlighted in green and cells with minimum values in yellow. FIE = arable field, FAL = fallow, GRO = grove, SRC1–SRC4 = SRC plots, HEA = headland, FOR = forest, and MEA = meadow. Table S12. Rounded averages of the species diversity indices Log series $\alpha$, Shannon exponential index (Shan-Expo), Shannon index, reciprocal Simpson index, and evenness for ground beetles, calculated from the values of five single traps per plot and year, and for the entire study period. The year of growth for the SRC plots is given in brackets. One-factorial ANOVA (Duncan test's, $p \leq 0.05$); degrees of freedom (df): 9.40; F = F-value; a–f indicate statistically significant differences between the plots. Cells with maximum values are highlighted in green and cells with minimum values in yellow. FIE = arable field, FAL = fallow, GRO = grove, SRC1–SRC4 = SRC plots, HEA = headland, FOR = forest, and MEA = meadow. Table S13. Rounded averages of the species diversity indices Log series $\alpha$, Shannon exponential index (Shan-Expo), Shannon index, reciprocal Simpson index, and evenness for the arachnids, calculated from the values of five single traps per plot and year, and for the entire study period. The year of growth for the SRC plots is given in brackets. One-factorial ANOVA (Duncan's test, $p \leq 0.05$); degrees of freedom (df): 9.40; F = F-value; a–g indicate statistically significant differences between the plots. Cells with maximum values are highlighted in green and cells with minimum values in yellow. FIE = arable field, FAL = fallow, GRO = grove, SRC1–SRC4 = SRC plots, HEA = headland, FOR = forest, and MEA = meadow. Table S14. Rounded averages of the habitat preference diversity indices Log series $\alpha$, Shannon exponential index (Shan-Expo), Shannon index, reciprocal Simpson index, and evenness for ground beetles, calculated from the values of five single traps per plot and year, and for the entire study period. The year of growth for the SRC plots is given in brackets. One-factorial ANOVA (Duncan's test, $p \leq 0.05$); degrees of freedom (df): 9.40; F = F-value; a–g indicate statistically significant differences between the plots. Cells with maximum values are highlighted in green and cells with minimum values in yellow. No. HP = mean number of different habitat preference groups. FIE = arable field, FAL = fallow, GRO = grove, SRC1–SRC4 = SRC plots, HEA = headland, FOR = forest, and MEA = meadow. Table S15. Rounded averages of the habitat preference diversity indices Log series $\alpha$, Shannon exponential index (Shan-Expo), Shannon index, reciprocal Simpson index, and evenness for the arachnids, calculated from the values of five single traps per plot and year, and for the entire study period. The year of growth for the SRC plots is given in brackets. One-factorial ANOVA (Duncan's test, $p \leq 0.05$); degrees of freedom (df): 9.40; F = F-value; a–h indicate statistically significant differences between the plots. Cells with maximum values are highlighted in green and cells with minimum values in yellow. No. HP = mean number of different habitat preference groups. FIE = arable field, FAL = fallow, GRO = grove, SRC1–SRC4 = SRC plots, HEA = headland, FOR = forest, and MEA = meadow. Table S16. Statistical parameters for the first four ordination axes of the redundancy analyses (RDAs) for ground beetles on the basis of summed individuals of five traps per plot in the different years of growth in the SRC and the reference plots, as well as for selected structural variables. VarExpl = variance explanation, F = F-statistic, $p$ = significance, uncorrected, padj = significance, corrected according to Holm, and Sign.adj = significance level: * $p \leq 0.05$, ** $p \leq 0.01$, *** $p \leq 0.001$, and n.s. = statistically not significant. Table S17. Statistical parameters for the first four ordination axes of the redundancy analyses (RDAs) for arachnids on the basis of summed individuals of five traps per plot in the different years of growth in the SRC and the reference plots, as well as for selected structural variables. VarExpl = variance explanation, F = F-statistic, $p$ = significance, uncorrected, padj = significance, corrected according to Holm, and Sign.adj = significance level: * $p \leq 0.05$, ** $p \leq 0.01$, *** $p \leq 0.001$, and n.s. = statistically not significant. Table S18. Correlation matrix of the structural variables (StrVar) for the first four ordination axes, which were calculated from the data of the structural variables (Axe E1–4) and the ground beetle species data (Axe R1–4). VIF = variance inflation factor. Redundancy analyses (RDAs) for the ground beetles in four years of growth in the SRC plots and the reference plots studied at the same time in the investigation period 2011–2014. The calculations were based on the Hellinger-transformed numbers of individuals from five single traps per plot and year of growth. Table S19. Correlation matrix of the structural variables (StrVar) for the first four ordination axes, which were calculated from the data of the structural variables (Axe E1–4) and the arachnid species data (Axe R1–4). VIF = variance inflation factor. Redundancy analyses (RDAs) for the arachnids

in four years of growth in the SRC plots and the reference plots studied at the same time in the investigation period 2011–2014. The calculations were based on the Hellinger-transformed numbers of individuals from five single traps per plot and year of growth. Table S20. Loadings (cases) of the plot points on the first four axes in the ordination diagram (triplot), which were calculated from the structural variables—environmental variables (CaseE)—and from the ground beetle species values—response variables (CaseR). Redundancy analyses (RDAs) in the entire study period 2011–2014. The calculations were based on the Hellinger-transformed numbers of individuals from five single traps per study plot. FIE = arable land, FAL = fallow, GRO = grove, SRC1–SRC4 = short-rotation coppice plots, HEA = headland, FOR = forest, and MEA = meadow. The numbers after the plot abbreviations indicate the years of investigation. Table S21. Loadings (cases) of the plot points on the first four axes in the ordination diagram (triplot), which were calculated from the structural variables—environmental variables (CaseE)—and from the arachnid species values—Response variables (CaseR). Redundancy analyses (RDAs) in the entire study period 2011–2014. The calculations were based on the Hellinger-transformed numbers of individuals from five single traps per study plot. FIE = arable land, FAL = fallow, GRO = grove, SRC1–SRC4 = short-rotation coppice plots, HEA = headland, FOR = forest, and MEA = meadow. The numbers after the plot abbreviations indicate the years of investigation. Tables S22 and S23. Mean values of percentages of individuals for different habitat preference (HP) groups of the ground beetle and arachnid cenoses in the plots SRC1–SRC4, based on the sums of individuals from five single traps per plot and year. One-factorial ANOVA (Duncan's test, $p \leq 0.05$); a–d indicate statistically significant differences between years. Maximum values are highlighted in green and minimum values in yellow. A timber harvest is marked with a vertical line. Degrees of freedom (df): 3.16. FIE = species of arable habitats, GRL = species of grassland habitats, RUD = species of ruderal habitats, DRY = species of open dry habitats, FOR = species of forest habitats, and MIX = species group with different habitat preferences. Table S24. Absolute numbers of individuals (Ind), dominance percentages (Dom), mean values ($\overline{x}$) and standard errors (SEs) of summerised habitat preference (HP) groups of ground beetle species in SRC1–SRC4 for the years 2011–2014. The year of growth is given in brackets for the SRC. Cells with maximum values are highlighted in green and cells with minimum values in yellow. FIE = species of arable fields, GRL = species of grassland habitats, DRY = species of open dry habitats, FOR = species of forest habitats, and MIX = species group with different habitat preferences. Table S25. Absolute numbers of individuals (Ind), dominance proportions (Dom) in percent, mean values ($\overline{x}$) and standard errors (SEs) of summerised habitat preference (HP) groups of the arachnid species in SRC1–SRC4 for the years 2011–2014. The year of growth is given in brackets for the SRC. Cells with maximum values are highlighted in green and cells with minimum values in yellow. FIE = species of arable fields, GRL = species of grassland habitats, RUD = species of ruderal habitats, FOR = species of forest habitats, and MIX = species group with different habitat preferences. Table S26. Annual total sum of individuals (Ind) and percentage of dominance (Dom) of stenotopic (steno) and eurytopic (eury) ground beetle species with a preference for different forest types in the study years 2011–2014. SRC = short-rotation coppice plots. For more detailed explanations, see Figure 11a–d. Table S27. Annual total sum of individuals (Ind) and percentage of dominance (Dom) of stenotopic (steno) and eurytopic (eury) arachnid species with a preference for different forest types in the study years 2011–2014. SRC = short-rotation coppice plots. For more detailed explanations, see Figure 11a–d. Table S28. Logarithmised (Log10) numbers of individuals of stenotopic and eurytopic forest species of ground beetles in SRC1–SRC4, calculated from five traps per study plot and year. One-factorial ANOVA (Duncan test $p \leq 0.05$), degrees of freedom (df): 9.24, a–c indicates statistically significant differences between the plots. Maximum values are highlighted in green, minimum values in yellow. For more detailed explanations, see Figure 11a–d. Table S29. Logarithmised (Log10) numbers of individuals of stenotopic and eury-topic forest species of arachnids SRC1–SRC4, calculated from five traps per study plot and year. One-factorial ANOVA (Duncan's test $p \leq 0.05$); degrees of freedom (df): 9.24; a–c indicate statistically significant differences between the plots. Maximum values are highlighted in green and minimum values in yellow. For more detailed explanations, see Figure 11a–d. Table S30. Statistical characteristics of the relationships between the proportions of individuals in % (ArcSin-root-transformed) of ecological types of ground beetles and selected structural variables. Linear multiple regression for the plots SRC1–SRC4 in the years 2011–2014. $R^2$ = coefficient of determination, SE = standard error, F = F-value, df = degrees of freedom, $p$ = significance, B = regression coefficient, and t = $t$-value. HAC = statistical parameters were calculated using robust Newey–West estimators. Variance inflation factor (VIF): shade: 1.379; herb: 2.955; grass: 1.956; litter: 4.392; open ground: 1.431. Table S31. Statistical characteristics of

the relationships between the proportions of individuals in % (ArcSin-root-transformed) of habitat preferences of ground beetles and selected structural variables. Linear multiple regression for the plots SRC1–SRC4 in the years 2011–2014. $R^2$ = coefficient of determination, SE = standard error, F = F-value, df = degrees of freedom, *p* = significance, B = regression coefficient, and t = *t*-value. HAC = statistical parameters were calculated using robust Newey–West estimators. Variance inflation factor (VIF): shade: 1.379; herb: 2.955; grass: 1.956; litter: 4.392; open ground: 1.431. Table S32. Statistical characteristics of the relationships between the proportions of individuals in % (ArcSin-root-transformed) of ecological types of arachnids and selected structural variables. Linear multiple regression for the plots SRC1–SRC4 in the years 2011–2014. $R^2$ = coefficient of determination, SE = standard error, F = F-value, df = degrees of freedom, *p* = significance, B = regression coefficient, and t = *t*-value. HAC = statistical parameters were calculated using robust Newey–West estimators. Variance inflation factor (VIF): shade: 1.379; herb: 2.955; grass: 1.956; litter: 4.392; open ground: 1.431. Table S33. Statistical characteristics of the relationships between the proportions of individuals in % (ArcSin-root-transformed) of habitat preferences of arachnids and selected structural variables. Linear multiple regression for the plots SRC1–SRC4 in the years 2011–2014. $R^2$ = coefficient of determination, SE = standard error, F = F-value, df = degrees of freedom, *p* = significance, B = regression coefficient, and t = *t*-value. HAC = statistical parameters were calculated using robust Newey–West estimators. Variance inflation factor (VIF): shade: 1.379; herb: 2.955; grass: 1.956; litter: 4.392; open ground: 1.431.

**Author Contributions:** J.K. and R.P.: Investigation, species identification, data analysis, writing—original draft preparation, review, and editing. M.G.: Writing—review and editing, project administration. All authors have read and agreed to the published version of the manuscript.

**Funding:** The investigations were conducted within the scope of the Federal Research Project 'Development of Extensive Land Use Concepts for the Production of Renewable Resources as Possible Compensation and Replacement Measures' (ELKE)–Phase III. The project was funded by the Federal Ministry of Food and Agriculture (BMEL) and the Agency for Renewable Resources (FNR) (Funding Code: FKZ: 220 077 09).

**Institutional Review Board Statement:** Not applicable.

**Data Availability Statement:** The original contributions presented in the study are included in the article/Supplementary Material, further inquiries can be directed to the corresponding author.

**Acknowledgments:** Our gratitude extends to the company Viessmann Werke GmbH & Co. KG, Allendorf/Eder for providing the research plots and the support of the project manager for biomass Hans-Moritz von Harling. We would also like to thank the farmers for their permission to conduct the studies on their land and the Viessmann and ZALF staff for their support in vegetation structure mapping and trap rotation.

**Conflicts of Interest:** The authors declare no conflicts of interest.

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
