# Peer review of "The Effects of Vegetation Structure and Timber Harvesting on Ground Beetle (Col.: Carabidae) and Arachnid Communities (Arach.: Araneae, Opiliones) in Short-Rotation Coppices"

_land, doi:10.3390/land13020145_

Round 1
Reviewer 1 Report
Comments and Suggestions for Authors 1. What is the main question addressed by the research? The authors analyse the relationship between insects and vegetation and introduce us to the results, which have theoretical implications for community ecology. 2. Do you consider the topic original or relevant in the field? The topic of the article is relevant. Does it address a specific gap in the field? The authors found a statistically significant correlation between the percentage coverage of structural variables and the quantitative and qualitative species composition in ground beetle and arachnid. 3. What does it add to the subject area compared with other published material? The paper contributes to the understanding of ecosystem relationships in agroforestry communities. 4. What specific improvements should the authors consider regarding the methodology? What further controls should be considered? In my opinion, everything is fine in this section. 5. Are the conclusions consistent with the evidence and arguments presented and do they address the main question posed? The authors have omitted the "Conclusion" section. This needs to be corrected and the main conclusions, theoretical and practical significance and any limitations of the application of the research results should be briefly stated. 6. Are the references appropriate? In my opinion, everything is fine in this section. 7. Please include any additional comments on the tables and figures. In my opinion, everything is fine in this section. The figures are of good quality, the table is informative, and there is an appendix that complements the actual material given in the main sections.I found the article very interesting. The size of the study attracts attention. The design of the study is well thought out and interesting. The topic of the article is relevant. The authors analyse the relationship between insects and vegetation and introduce us to the results, which have theoretical implications for community ecology.
I would like to draw your attention to the fact that the title of the paper is too long. I recommend that the authors improve this, the wording should be concise and clear.
The abstract can indicate the practical and theoretical significance of the research results.
The relevance of the study is well argued in the introduction. The research objectives are clearly stated.
The methodology approaches are described in detail. The authors used methods adequate to the tasks set.
The research results are illustrated with figures and tables that are informative and do not duplicate each other. The paper contains 1 informative table and 15 visual figures. The results are presented clearly and clearly.
The authors have omitted the "Conclusion" section. This needs to be corrected and the main conclusions, theoretical and practical significance and any limitations of the application of the research results should be briefly stated.
The paper will be of interest to a wide range of readers whose scientific interests are related to community ecology. Despite the fact that English is not my native language, I read the paper with interest and had no difficulties in understanding. The paper corresponds to the subject of Land.
Author Response
Dear Reviewer 1,
We would like to thank you very much for your conscientious review of our manuscript. Your suggestions for improvement were very valuable and helpful to us.
I would like to draw your attention to the fact that the title of the paper is too long. I recommend that the authors improve this, the wording should be concise and clear.
Response: On your advice, we shortened the title of our manuscript. You were absolutely right, the title was too long.
The authors have omitted the "Conclusion" section. This needs to be corrected and the main conclusions, theoretical and practical significance and any limitations of the application of the research results should be briefly stated.
Response: Thank you very much for drawing our attention to this important aspect. We have added the conclusions in lines 837-857.
We have also added a categorization of the significance of our results in lines 827-834 of the Discussion.
Sincerely,
Jessika Konrad, Ralph Platen and Michael Glemnitz
Reviewer 2 Report
Comments and Suggestions for Authors
In the paper “The Influence of Vegetation Structure and the Effects of Timber Harvest in Short Rotation Coppices (SRCs) on the Diversity and Structure of Ground Beetle (Col.: Carabidae) and Arachnid Communities (Arach.: Araneae, Opiliones)” the authors investigated the impact of vegetation structure on the composition and diversity of ground beetle and arachnid communities in four SRC and six reference plots from 2011 to 2014 in an agricultural landscape in Hesse, Germany. This manuscript is well organized, and the drawn conclusions are coherent with the obtained results. I have enjoyed reading your paper; however, the manuscript must be revised by an English native speaker to fix several grammatical errors that I detected in the paper. I hope to provide very useful suggestions to improve the overall clarity of your study as well as the quality of your analysis. I think that my suggestions look feasible to you, and I believe you will be able to address them. Thus, please take care to do a full revision of your manuscript according to all my comments. Improvements based on my comments will be crucial for acceptance. I have some concerns and suggestions for each aspect of the manuscript. Please see below.
Abstract: You need to highlight more your results. Furthermore, add a last sentence on the conclusions of your study.
Introduction: The paper is technically sound and the claims are convincing. However I think that some references should be updated. Please, note that the hypothesis and the predictions are unclear, you need to well explain them.
Lines 42 – 43: I think that you should add these important references as examples to support your sentence: “However, the decline in insect diversity is not solely limited to agricultural areas but has also been observed, among other places, in forests”. I would like to suggest:
Russo, D., et al., (2015). Protecting one, protecting both? Scale‐dependent ecological differences in two species using dead trees, the rosalia longicorn beetle and the barbastelle bat. Journal of Zoology, 297(3), 165-175.
Wagner, D. L., et al., (2021). A window to the world of global insect declines: Moth biodiversity trends are complex and heterogeneous. Proceedings of the National Academy of Sciences, 118(2), e2002549117.
Lines 53 – 55: I think that you should add these important references as examples to support your sentence: “Especially for species that reliant on various habitats during their life cycle, the close proximity and accessibility of different habitats for development, foraging, hibernation, and reproduction are crucial”. I would like to suggest:
Ogilvie, J. E., & CaraDonna, P. J. (2022). The shifting importance of abiotic and biotic factors across the life cycles of wild pollinators. Journal of Animal Ecology, 91(12), 2412-2423.
Lines 68 – 69: I think that you should add these important references as examples to support your sentence: “Another aspect involves landscape fragmentation and the resulting degree of 68 isolation among habitats”.
Lines 85 – 114: Please, reduce this part of the manuscript.
Chetcuti, J., et al., (2020). Habitat fragmentation increases overall richness, but not of habitat-dependent species. Frontiers in Ecology and Evolution, 8, 607619.
Lines 116 – 134: Please, explain in detail you hypothesis and predictions.
Materials and methods: In general, the methods are appropriate and the study seems well conducted, although some details deserve a bit more attention i.e., especially about the methodology and the data. All the script used in this paper must be added in the supplementary materials. Please, provide also all the link to source where you downloaded the data.
Line 144: Please, add the north symbol and the scale in the map.
Results: Well written! The figures and the tables are all informative and necessary, but not redundant, ensuring the correct comprehension of the manuscript.
Line 325: What is the label on the y-axis? Then, Are you sure to detect values more than 100%?
Line 462: Please, note that in the results you cannot add references. Please, delete all the references from this section of the manuscript.
Discussion: The paper discussed appropriately the context and the theme, although there is important literature not cited by the authors. I think that the authors should be discussing their results also comparing them with those already published on other species/genus/family. In fact your paper discusses findings in relation to some of the work in the field but ignores other important work that I think should be added in your discussion. The discussion contains some irrelevant information and repeats results without interpretation. Please, pay more attention to this last point.
Comments on the Quality of English LanguageI have enjoyed reading your paper; however, the manuscript must be revised by an English native speaker to fix several grammatical errors that I detected in the paper.
Author Response
Dear Reviewer 2,
We would like to thank you very much for taking the time to thoroughly review our manuscript. Your suggestions for improvement were very valuable and helpful to us. We have considered your comments in detail and have endeavored to incorporate them conscientiously.
In order to improve the readability of the manuscript, we have decided to use an English-language editing service.
Below you will find detailed documentation on the implementation of your individual suggestions for improvement.
Abstract: You need to highlight more your results. Furthermore, add a last sentence on the conclusions of your study.
Response: Thank you for drawing our attention to these important aspects. We have emphasized the results more strongly in lines 20-28 and added a concluding sentence to our conclusions in line 29.
Introduction: I think that some references should be updated. Please, note that the hypothesis and the predictions are unclear, you need to well explain them.
Response: Thank you for drawing our attention to this important point. We have revised this section in lines 108-125 and tried to make our research questions and hypotheses more precise.
Lines 42 – 43: I think that you should add these important references as examples to support your sentence: “However, the decline in insect diversity is not solely limited to agricultural areas but has also been observed, among other places, in forests”. I would like to suggest:
Russo, D., et al., (2015). Protecting one, protecting both? Scale‐dependent ecological differences in two species using dead trees, the rosalia longicorn beetle and the barbastelle bat. Journal of Zoology, 297(3), 165-175.
Wagner, D. L., et al., (2021). A window to the world of global insect declines: Moth biodiversity trends are complex and heterogeneous. Proceedings of the National Academy of Sciences, 118(2), e2002549117.
Response: Thank you for your reference to these relevant publications on this topic. We have added your suggestions in line 43.
Lines 53 – 55: I think that you should add these important references as examples to support your sentence: “Especially for species that reliant on various habitats during their life cycle, the close proximity and accessibility of different habitats for development, foraging, hibernation, and reproduction are crucial”. I would like to suggest:
Ogilvie, J. E., & CaraDonna, P. J. (2022). The shifting importance of abiotic and biotic factors across the life cycles of wild pollinators. Journal of Animal Ecology, 91(12), 2412-2423.
Response: We regret that we could not include this highly interesting publication here, because we do not consider the bumblebee species studied to be suitable examples of species that depend on several different habitats during their life cycle.
Lines 68 – 69: I think that you should add these important references as examples to support your sentence: “Another aspect involves landscape fragmentation and the resulting degree of 68 isolation among habitats”.
Chetcuti, J., et al., (2020). Habitat fragmentation increases overall richness, but not of habitat-dependent species. Frontiers in Ecology and Evolution, 8, 607619.
Response: This publication is a good addition. We have inserted this citation in line 69.
Lines 85 – 114: Please, reduce this part of the manuscript.
Response: You are absolutely right. We have removed this section.
Lines 116 – 134: Please, explain in detail you hypothesis and predictions.
Response: Thank you for drawing our attention to this important aspect. We have revised and clarified this section in lines 108-125.
Materials and Methods: In general, the methods are appropriate and the study seems well conducted, although some details deserve a bit more attention i.e., especially about the methodology and the data. All the script used in this paper must be added in the supplementary materials. Please, provide also all the link to source where you downloaded the data.
Response: We did not use data from other sources. We only analyzed data that we collected ourselves during the field study. Unfortunately, the software we used does not provide any code. Therefore, we limited ourselves to explaining the settings used in the tests of the analyses we conducted.
Line 144: Please, add the north symbol and the scale in the map.
Response: The north sign and the scale were added to the map.
Results: Line 325: What is the label on the y-axis? Then, Are you sure to detect values more than 100%?
Response: The label on the y-axis has been added. Yes. The visual estimation of the percent cover of vegetation was done "two-dimensionally" based on 100% per plot. The relative shading was estimated with the help of an imaginary additional "layer", resulting in a total cover of > 100 %.
Line 462: Please, note that in the results you cannot add references. Please, delete all the references from this section of the manuscript.
Response: Thank you for pointing out this important point. This reference has been removed and moved to the method section after lines 278-279. The results section has been checked and no further references are included in this section.
Discussion: The paper discussed appropriately the context and the theme, although there is important literature not cited by the authors. I think that the authors should be discussing their results also comparing them with those already published on other species/genus/family. In fact your paper discusses findings in relation to some of the work in the field but ignores other important work that I think should be added in your discussion. The discussion contains some irrelevant information and repeats results without interpretation. Please, pay more attention to this last point.
Response: Our investigations are limited to the analyses of ground beetle and arachnid cenoses. Other groups of animals show considerable differences in their ecology, their habitat requirements, their reactions to changes in their habitat and also in their characteristics compared to ground beetles and arachnids. In addition, to our knowledge, there are no studies in SRCs that are comparable to our experimental design. To consider these aspects in detail would considerably expand the scope of the discussion and distract from our focus. We therefore decided against including this recommendation and hope for your understanding.
A missing interpretation has been added in lines 689-690.
I have enjoyed reading your paper; however, the manuscript must be revised by an English native speaker to fix several grammatical errors that I detected in the paper.
Response: We decided to use an English-language editing service to improve the readability of the manuscript.
Sincerely,
Jessika Konrad, Ralph Platen and Michael Glemnitz
Reviewer 3 Report
Comments and Suggestions for Authors
Dear Authors
This manuscript investigates the diversity of ground beetles and arachnids in short rotation coppices (SRCs) compared to reference plots. It presents interesting findings, but requires significant improvement before publication.
Strengths:
- Interesting topic: The research addresses a relevant and timely topic of insect diversity in agricultural landscapes.
- Comprehensive methods: The methodology utilizes standardized sampling and analysis techniques, demonstrating rigor.
- Detailed data: The results section provides rich data on species diversity, ecological traits, and habitat preferences.
Areas for Improvement:
- Writing clarity and conciseness: Sentences are often long and complex, hindering readability. Consider shorter, clearer sentence structures and simpler word choice.
- Grammatical errors: Numerous grammatical errors and stylistic inconsistencies affect the flow and professional appearance of the manuscript. Thorough proofreading is necessary.
- Specific details: Some descriptions lack specificity, especially regarding the justification for choosing specific methodologies and interpretations of results.
- Discussion depth: While the discussion touches on ecological implications, it could benefit from stronger elaboration and supporting evidence for key findings.
- Novelty and significance: The manuscript would be strengthened by emphasizing the novelty and potential impact of the findings for conservation and management practices.
- Conclusion: A concise and impactful conclusion summarizing the key takeaways is missing.
Overall:
This manuscript has the potential for publication but requires significant revisions to address the identified weaknesses. Focusing on grammatical clarity, conciseness, specific details, deeper discussion, and a strong conclusion will significantly improve the quality and impact of the research.
Recommendations:
- Revise the writing style for clarity and conciseness.
- Carefully proofread and address all grammatical errors.
- Provide more specific details and justifications in the methods and discussion sections.
- Deepen the discussion by elaborating on ecological implications and supporting evidence.
- Emphasize the novelty and potential impact of the findings.
- Add a concise and impactful conclusion summarizing the key takeaways.
With thorough revisions, this manuscript can become a valuable contribution to the field of insect biodiversity in agricultural landscapes.
I hope this feedback is helpful!
Best regards
Specific recommendations for improvement:
***Title:
General writing errors
- The title is too long. A good title should be concise and easy to remember.
- The title is not specific enough. It does not provide enough information about the study's methods or findings.
Overall assessment
As a reviewer, I would recommend that the author revise the title as follows:
Title: The effects of vegetation structure and timber harvest on ground beetle and arachnid communities in short rotation coppices
This title is shorter and more concise. It also provides more specific information about the study's methods and findings.
Here are some additional suggestions for improving the title:
- The author could add the name of the region or country where the study was conducted.
- The author could add the name of the specific species of ground beetles and arachnids that were studied.
By making these revisions, the author could create a title that is more informative and engaging.
***Abstract:
Grammatical and scientific errors in the abstract:
Landscape complexity: While "landscape complexity" is understandable, consider replacing it with more specific terms like "habitat heterogeneity" or "structural diversity" for clarity.
High structural heterogeneity: Instead of repeating "heterogeneity," use synonyms like "variability" or "diversity" within the sentence.
Statistically significant correlations: Redundant phrasing. Simply say "correlations existed" or "were found."
Proportion of individuals: Repetitive. Consider condensing to "the number of individuals of forest species..."
Open-land species: Similar to above, consider "species found in open landscapes."
Species- and Habitat-Preference Diversity: These terms might not be familiar to a general audience. Briefly define them within the abstract or use simpler alternatives like "species diversity" and "habitat specialization."
Ecological equivalence: This needs clarification for non-specialists. Consider saying "similar ecological roles" or "functional redundancy."
General writing errors:
Sentence length: Some sentences are long and complex. Break them down into shorter, more digestible units.
Specificity: The abstract could benefit from more specific details about the study, such as the type of crops in the agricultural landscape or the species of ground beetles and arachnids studied.
Focus: The abstract mentions both timber harvest and cenotic diversity, but it's unclear which aspect is the main focus of the study. Prioritize the key finding in the abstract.
Overall assessment:
The abstract is well-written but could be improved with more concise and specific language, avoiding redundancy and technical jargon where possible. Consider revising the abstract to:
Be more concise and avoid repetition.
Use simpler language for a broader audience.
Clearly state the main finding and focus of the study.
Provide more specific details about the study methods and context.
By making these revisions, you can create a more impactful and informative abstract that effectively summarizes your research.
Additional notes:
As a reviewer, I would encourage the author to consider the target audience for this abstract. If it is intended for a scientific journal, the current level of technical language may be appropriate. However, if it is intended for a wider audience, such as policymakers or the general public, then simpler language should be used.
I would also recommend that the author carefully proofread the abstract for any remaining grammatical errors.
I hope this feedback is helpful!
***Introduction
Grammatical and scientific errors in the introduction:
- Repetition: There are some instances of repetition that could be condensed for improved readability. For example, "insect diversity decline" can be replaced with "insect decline" in some cases.
- Sentence length: Some sentences are quite long and could be broken down into shorter units for easier comprehension.
- Specificity: While the introduction mentions various factors affecting insect diversity, it could benefit from mentioning specific examples relevant to the study region or species being investigated.
- Citations: Ensure all citations are formatted correctly and consistently according to the journal's style guide.
General writing errors:
- Focus: The introduction could be more focused on the specific research question and hypotheses of the study. While providing context is important, prioritize the core objectives of your research.
- Flow: The transition between different sections of the introduction could be smoother. Consider using clear transitions to guide the reader from the general context to the specific aims of the study.
Overall assessment:
The introduction provides a good overview of the research topic and highlights the importance of SRCs in promoting insect diversity. However, it could be improved by addressing the points mentioned above to enhance clarity, conciseness, and focus on the specific research question.
Here are some specific suggestions for improvement:
- Condense repetitive phrases and sentences.
- Break down long sentences into shorter units.
- Provide more specific examples of insect diversity decline and its drivers relevant to your study region or species.
- Clearly state the research question and hypotheses of the study.
- Improve the flow between different sections of the introduction.
- Ensure all citations are formatted correctly.
By making these revisions, you can create a stronger and more impactful introduction that effectively sets the stage for your research.
Additional notes:
- The introduction provides a good justification for choosing ground beetles and arachnids as the study organisms.
- The mention of different research gaps and limitations related to previous studies on SRCs demonstrates a good understanding of the field.
- Overall, the introduction is well-written and informative, but it could be further strengthened by addressing the points mentioned above.
Grammatical errors in the introduction:
Repetition:
"In general, species can only successfully reproduce when they encounter suitable habitats. Hence, the loss of habitats holds particular significance in species decline [6]."
These two sentences are repetitive and could be replaced with one sentence. For example: "In general, the loss of suitable habitats is a major driver of species decline."
"Short Rotation Coppices (SRCs), characterized by their spatially and temporally high structural heterogeneity, can contribute to diversifying agricultural landscapes."
This sentence is also repetitive and could be replaced with one sentence. For example: "SRCs, with their high structural heterogeneity, can contribute to diversifying agricultural landscapes."
Sentence length:
Some sentences in the introduction are very long and could be broken down into shorter units for easier comprehension. For example:
"Numerous studies address changes in insect diversity within the agricultural landscape and their causes. Land-use change [1-4], agricultural intensification [2,5], high levels of pesticide application [6,7], narrow crop rotations [8,9], and the absence of landscape complexity [10-12] are widely considered major contributors to the decline in insect diversity."
This could be broken down into the following sentences:
Numerous studies have addressed changes in insect diversity within the agricultural landscape and their causes.
Land-use change, agricultural intensification, high levels of pesticide application, narrow crop rotations, and the absence of landscape complexity are all considered major contributors to the decline in insect diversity.
Specificity:
Some sentences in the introduction lack specificity and could be improved by providing more specific examples. For example:
"The fact that more than half of Germany's land area was under agricultural use in 2022 [13] underscores the importance of agriculture in preserving species diversity."
This could be improved by providing more specific information about the types of land use in Germany. For example:
The fact that more than half of Germany's land area was under agricultural use in 2022, including arable land, grassland, and forests, underscores the importance of agriculture in preserving species diversity.
I hope this feedback is helpful!
*** Materials and Methods
Assessment of the Materials and Methods Section
Grammar and Clarity:
- Overall, the section is well-written and grammatically correct. However, there are a few minor typos and errors in punctuation that could be improved. For example:
- Line 185: "The percentage coverage of shading from trees (shade), shrubs, crops, grass, herbaceous plants, moss, litter layer, deadwood (≥ 2 cm to ≤ 10 cm and ≥ 10 cm diameter), as well as vegetation-free, open ground," Consider breaking this long list into two sentences for easier reading.
- Some sentences could be made more concise by removing unnecessary words or phrases. For example:
- Line 175: "The Arable Field (FIE) was conventionally cultivated and planted with winter cereals from 2011 to 2013, followed by summer cereals in 2014." This could be shortened to: "FIE was planted with winter cereals (2011-2013) and summer cereals (2014)."
Scientific and Technical Accuracy:
- The methods described are comprehensive and appropriate for the study. The use of standardized sampling techniques and data analysis methods demonstrates methodological rigor.
- The section clearly defines the ecological traits and habitat preferences used to categorize ground beetles and arachnids.
- The justification for using different diversity indices for different types of data is clear and well-explained.
Principles and Rules of Scientific Writing:
- The section is well-organized and easy to follow. The subsections are logical and provide a clear roadmap for the research methodology.
- The methods are described in sufficient detail to allow for repeatability by other researchers.
- The use of references is appropriate and provides the reader with access to the source material for further details.
Specific Recommendations:
- Consider providing brief justifications for choosing specific pitfall trap design, preservation solution, and trapping duration.
- Clarify whether the differentiation of ETs and HPs was performed separately for ground beetles and arachnids or combined for the analysis.
- Explain the rationale behind combining spiders and harvestmen into a single "arachnid" category for analysis.
Overall Impression:
The Materials and Methods section is well-written and provides a clear and detailed description of the research methodology. With a few minor revisions, it would be ready for publication in a scientific journal.
Additional Comments:
- As a reviewer, I would likely ask for some clarification on the points mentioned in the Specific Recommendations above.
- I would also be interested in knowing more about the rationale for choosing SRCs as the study object and the specific research questions the study is addressing.
Please note that this is just one reviewer's perspective. Other reviewers may have different concerns or questions.
I hope this feedback is helpful!
****Results
Grammatical errors and stylistic issues:
Sentence structure:
- Sentences are often long and complex, making them difficult to read. Aim for shorter, clearer sentences.
- Subject-verb agreement errors: Ensure subject-verb agreement in all sentences.
- Redundancy: Eliminate redundant phrases and information.
- Parallelism: Use parallel sentence structure when comparing or listing things.
Examples:
- "The vegetation structure in the SRCs consisted of the variables ‘herbage’, ‘grass’, ‘litter’ and ‘open’ (Figure 3). Additionally, these plots were notably characterized by predominantly high shading levels (Shade). Considerable differences in the coverage levels of these variables were characteristic in the SRCs across the years, whereas this was observed to a lesser extent in the reference plots (Tables A7 and A8)." (Combine the two sentences and rephrase for clarity.)
- "Across the entire study period, the ground beetle cenoses in the SRCs, except for SRC4, exhibited statistically significantly higher Shannon indices for Species-Diversity compared to the reference plots (Figure 6a)." (Simplify sentence structure and word choice.)
Word choice:
- Some words are not used correctly or could be replaced with more precise alternatives.
- Use less jargon and technical terms, especially when writing for a general audience.
Examples:
- "Cenoses" instead of "communities" might be confusing for a general audience.
- "Considerable differences" could be replaced with a more specific term like "substantial variations" or "marked fluctuations."
Clarity and conciseness:
- The writing can be more concise and impactful. Avoid unnecessary jargon and technical terms.
- Ensure smooth flow between sentences and paragraphs.
Examples:
- "In total, between 2011 and 2014, across all study plots, 103 ground beetle species with 46,617 individuals and 181 spider species with 42,252 individuals (167 spider species with 35,623 individuals and 14 harvestman species with 6,629 individuals) were recorded (Tables A1 and A2)." (This sentence can be broken down into several shorter sentences for better clarity.)
**** Discussion
Grammatical and stylistic issues:
- Sentence structure: Sentences are often long and complex, making them difficult to read. Aim for shorter, clearer sentences.
- Word choice: Some words are not used correctly or could be replaced with more precise alternatives. Avoid jargon and technical terms for a wider audience.
- Redundancy: Eliminate unnecessary phrases and redundancies.
- Conciseness: The writing can be more concise and impactful. Avoid padding and overly technical terminology.
- Clarity and flow: Some transitions between sentences and paragraphs could be smoother. Ensure consistent clarity and logical flow throughout the discussion.
- Citations and references: Ensure proper formatting and consistent citation of references throughout the article.
Scientific, technical, and article writing principles:
- Species names: Latin species names are missing for many animal groups mentioned.
- Statistical terminology: The use of some statistical terms (e.g., "highly significant") could be replaced with more specific p-values or confidence intervals.
- Visual aids: Consider incorporating figures or tables to present data more effectively.
- Discussion depth: Some statements need further elaboration or supporting evidence.
- Novelty and significance: The novelty and potential impact of the findings could be emphasized more.
- Conclusion: While the discussion touches on ecological implications, a stronger and more concise conclusion summarizing the key takeaways would be beneficial.
Overall evaluation:
The discussion section presents interesting findings about ground beetle and arachnid communities in SRCs. However, it would benefit significantly from improvement in the identified areas above. Specifically, focusing on grammatical clarity, conciseness, precise word choice, and incorporating specific statistical results would improve the readability and scientific rigor of the text. Additionally, addressing the points concerning novelty, significance, and a clear conclusion would strengthen the overall impact of the discussion.
Recommendation: Consider revising the discussion section, addressing the listed grammatical, stylistic, and scientific issues. Strengthening the clarity, conciseness, and focus on key findings would significantly improve the overall quality and impact of the research.
Specific examples of grammatical errors
In addition to the general grammatical issues mentioned above, there are a number of specific grammatical errors in the discussion section. For example:
- The word "than" is used incorrectly in the following sentence:
Original: The mean proportions of arable and grassland species in both animal groups exhibit statistically significantly lower values in the oldest growth year compared to the youngest.
Revised: The mean proportions of arable and grassland species in both animal groups are statistically significantly lower in the oldest growth year than in the youngest.
- The word "particularly" is used incorrectly in the following sentence:
Original: The Species- and Habitat-Preference Diversity within the SRCs is statistically significantly higher in both animal groups compared to reference plots. Particularly in highly dynamic habitats like SRC, the presence of species with various habitat preferences can be advantageous.
Revised: The Species- and Habitat-Preference Diversity within the SRCs is statistically significantly higher in both animal groups compared to reference plots. In highly dynamic habitats like SRC, the presence of species with various habitat preferences can be advantageous.
- The word "insurmountable" is used incorrectly in the following sentence:
Original: This suggests the potential of rotationally harvested SRC as a connecting element between wooded habitats in agricultural landscapes within surmountable distances.
Revised: This suggests the potential of rotationally harvested SRC as a connecting element between wooded habitats in agricultural landscapes within reasonable distances.
**** Conclusion: missing
Comments on the Quality of English Language
mentioned in the previous box.
Author Response
Dear Reviewer 3,
We would like to thank you very much for taking the time to thoroughly review our manuscript. Your extensive suggestions for improvement were very valuable and helpful to us. We have taken your comments into account in detail and hope that we have been able to improve the quality of our manuscript as a result.
In order to improve the readability of the manuscript, we have decided to use an English-language editing service.
Below you will find detailed documentation on the implementation of your individual suggestions for improvement.
Title:
General writing errors
The title is too long. A good title should be concise and easy to remember.
The title is not specific enough. It does not provide enough information about the study's methods or findings.
Overall assessment
As a reviewer, I would recommend that the author revise the title as follows:
Title: The effects of vegetation structure and timber harvest on ground beetle and arachnid communities in short rotation coppices
Response: Thank you very much for your comment and your very concise suggestion, which we have gladly adopted. You were absolutely right, the title was too long.
Abstract:
Grammatical and scientific errors in the abstract:
Landscape complexity: While "landscape complexity" is understandable, consider replacing it with more specific terms like "habitat heterogeneity" or "structural diversity" for clarity.
Response: The phrase "high structural heterogeneity" was clarified in line 14 with "habitat heterogeneity".
High structural heterogeneity: Instead of repeating "heterogeneity," use synonyms like "variability" or "diversity" within the sentence.
Response: We have avoided repetition by rewording.
Statistically significant correlations: Redundant phrasing. Simply say "correlations existed" or "were found."
Response: The redundant wording has been replaced by "correlations" (line 20).
Proportion of individuals: Repetitive. Consider condensing to "the number of individuals of forest species..."
Response: The statement in line 22 does not refer to absolute numbers of individuals, but to relative proportions of individuals and could therefore only be modified.
Open-land species: Similar to above, consider "species found in open landscapes."
Response: see above.
Species- and Habitat-Preference Diversity: These terms might not be familiar to a general audience. Briefly define them within the abstract or use simpler alternatives like "species diversity" and "habitat specialization."
Response: Thank you very much for pointing this out. We have replaced these terms in lines 24-27 with explanatory descriptions.
Ecological equivalence: This needs clarification for non-specialists. Consider saying "similar ecological roles" or "functional redundancy."
Response: The term "ecological equivalence" has been replaced by "functional redundancy" in lines 26,27.
Response: We would like to thank you very much for your suggestions for alternative wording. These were extremely helpful to us.
General writing errors:
Sentence length: Some sentences are long and complex. Break them down into shorter, more digestible units.
Response: We have shortened the sentences and hope that they are now easier to understand.
Specificity: The abstract could benefit from more specific details about the study, such as the type of crops in the agricultural landscape or the species of ground beetles and arachnids studied.
Response: Unfortunately, we were unable to add this due to the limited scope of the abstract.
Focus: The abstract mentions both timber harvest and cenotic diversity, but it's unclear which aspect is the main focus of the study. Prioritize the key finding in the abstract.
Response: The abstract has been fundamentally revised. We hope to have focused and prioritized the individual aspects more strongly (lines 20-28).
Overall assessment:
Be more concise and avoid repetition.
Response: We have shortened the sentences and avoided repetition by rephrasing.
Use simpler language for a broader audience.
Response: Thank you for your comment. We have replaced several technical terms with explanatory descriptions in line with your detailed comments.
Clearly state the main finding and focus of the study.
Response: We have revised this section. In lines 20-28 we clarify the main finding and the focus of the study.
Provide more specific details about the study methods and context.
Response: Based on your suggestions, we have tried to explain the context and methodological details more concisely by rewording. Unfortunately, due to the limited scope of the abstract, we were unable to include additional details.
By making these revisions, you can create a more impactful and informative abstract that effectively summarizes your research.
Response: We hope to have achieved this through a fundamental revision of the abstract, taking into account your suggestions for improvement.
Additional notes:
As a reviewer, I would encourage the author to consider the target audience for this abstract. If it is intended for a scientific journal, the current level of technical language may be appropriate. However, if it is intended for a wider audience, such as policymakers or the general public, then simpler language should be used.
Response: We agree with this comment. In fact, our manuscript is aimed at the interested readership of a scientific journal. Notwithstanding this, we were happy to implement your suggestions regarding simpler language and the avoidance of technical terms in the abstract.
I would also recommend that the author carefully proofread the abstract for any remaining grammatical errors.
Response: We decided to use an English-language editing service to improve the readability of the manuscript.
Introduction
Grammatical and scientific errors in the introduction:
Repetition: There are some instances of repetition that could be condensed for improved readability. For example, "insect diversity decline" can be replaced with "insect decline" in some cases.
Response: The phrase "decline in insect diversity" has been replaced with "insect decline" in lines 42, 50 and 54.
Sentence length: Some sentences are quite long and could be broken down into shorter units for easier comprehension.
Response: Thank you very much for pointing out this aspect. We have split the long sentence constructions into shorter sentences.
Specificity: While the introduction mentions various factors affecting insect diversity, it could benefit from mentioning specific examples relevant to the study region or species being investigated.
Response: We agree with this comment. We have therefore inserted the results of a 10-year study in lines 49-53.
Citations: Ensure all citations are formatted correctly and consistently according to the journal's style guide.
Response: This has been carefully checked and corrected.
General writing errors:
Focus: The introduction could be more focused on the specific research question and hypotheses of the study. While providing context is important, prioritize the core objectives of your research.
Response: Thank you for pointing out this important point. We have revised and clarified this section in lines 108-125.
Flow: The transition between different sections of the introduction could be smoother. Consider using clear transitions to guide the reader from the general context to the specific aims of the study.
Response: The transitions between the individual sections of the introduction have been revised.
Here are some specific suggestions for improvement:
Condense repetitive phrases and sentences.
Response: To avoid repetition, we have reworded the relevant sentences (see specific notes)
Break down long sentences into shorter units.
Response: We have divided the long sentence constructions into shorter sentences.
Provide more specific examples of insect diversity decline and its drivers relevant to your study region or species.
Response: We have included the results of a 10-year study in lines 49-53.
Clearly state the research question and hypotheses of the study.
Response: We have revised this section in lines 108-125 to make our research questions and hypotheses clearer.
Improve the flow between different sections of the introduction.
Response: The transitions between the individual sections of the introduction have been harmonized.
Ensure all citations are formatted correctly.
Response: This has been carefully checked and corrected.
Grammatical errors in the introduction:
Repetition:
"In general, species can only successfully reproduce when they encounter suitable habitats. Hence, the loss of habitats holds particular significance in species decline [6]."
These two sentences are repetitive and could be replaced with one sentence. For example: "In general, the loss of suitable habitats is a major driver of species decline."
Response: You are absolutely right. We have replaced these two sentences in line 54 with your suggestion.
"Short Rotation Coppices (SRCs), characterized by their spatially and temporally high structural heterogeneity, can contribute to diversifying agricultural landscapes."
This sentence is also repetitive and could be replaced with one sentence. For example: "SRCs, with their high structural heterogeneity, can contribute to diversifying agricultural landscapes."
Response: This sentence has been shortened to the following wording in lines 112, 113: "Compared to traditional agricultural biotopes, SRC are characterized by a higher structural diversity."
Sentence length:
Some sentences in the introduction are very long and could be broken down into shorter units for easier comprehension. For example:
"Numerous studies address changes in insect diversity within the agricultural landscape and their causes. Land-use change [1-4], agricultural intensification [2,5], high levels of pesticide application [6,7], narrow crop rotations [8,9], and the absence of landscape complexity [10-12] are widely considered major contributors to the decline in insect diversity."
This could be broken down into the following sentences:
Numerous studies have addressed changes in insect diversity within the agricultural landscape and their causes.
Response: We have gratefully adopted your proposed amendment in line 36.
Land-use change, agricultural intensification, high levels of pesticide application, narrow crop rotations, and the absence of landscape complexity are all considered major contributors to the decline in insect diversity.
Response: We have also inserted this proposal in line 39.
Specificity:
Some sentences in the introduction lack specificity and could be improved by providing more specific examples. For example:
"The fact that more than half of Germany's land area was under agricultural use in 2022 [13] underscores the importance of agriculture in preserving species diversity."
This could be improved by providing more specific information about the types of land use in Germany. For example:
The fact that more than half of Germany's land area was under agricultural use in 2022, including arable land, grassland, and forests, underscores the importance of agriculture in preserving species diversity.
Response: We have clarified the wording "the area of Germany" by "the total area of Germany" in line 40.
I hope this feedback is helpful!
Response: Yes, your feedback was extremely helpful to us!
Materials and Methods
Grammar and Clarity:
Line 185: "The percentage coverage of shading from trees (shade), shrubs, crops, grass, herbaceous plants, moss, litter layer, deadwood (≥ 2 cm to ≤ 10 cm and ≥ 10 cm diameter), as well as vegetation-free, open ground," Consider breaking this long list into two sentences for easier reading.
Response: This sentence has been split into two sentences to improve readability (lines 187-190).
Some sentences could be made more concise by removing unnecessary words or phrases. For example:
Line 175: "The Arable Field (FIE) was conventionally cultivated and planted with winter cereals from 2011 to 2013, followed by summer cereals in 2014." This could be shortened to: "FIE was planted with winter cereals (2011-2013) and summer cereals (2014)."
Response: Thank you very much for your shortened formulation. We have modified your suggestion and inserted it in lines 177-178. We consider the mention of conventional farming to be relevant for differentiation from organic farming.
Specific Recommendations:
Consider providing brief justifications for choosing specific pitfall trap design, preservation solution, and trapping duration.
Response: Corresponding explanations have been added in lines 175-177, 195-197 and 199-200.
Clarify whether the differentiation of ETs and HPs was performed separately for ground beetles and arachnids or combined for the analysis.
Response: In lines 224 and 236, the differentiation "per animal group" and "for each animal group" has been added.
Explain the rationale behind combining spiders and harvestmen into a single "arachnid" category for analysis.
Response: This is explained in lines 210-211.
Additional Comments:
As a reviewer, I would likely ask for some clarification on the points mentioned in the Specific Recommendations above.
I would also be interested in knowing more about the rationale for choosing SRCs as the study object and the specific research questions the study is addressing.
Please note that this is just one reviewer's perspective. Other reviewers may have different concerns or questions.
Response: In the introduction, the section on the formulation of the hypotheses has been revised and the choice of SRCs as the object of investigation and the specific research questions explained (lines 108-125). We hope that this clarifies our underlying considerations.
Results
Grammatical errors and stylistic issues:
Sentence structure: Sentences are often long and complex, making them difficult to read. Aim for shorter, clearer sentences.
Response: You are right. We have shortened the long sentences for better understanding.
Subject-verb agreement errors: Ensure subject-verb agreement in all sentences.
Response: We decided to use an English-language editing service to ensure this.
Redundancy: Eliminate redundant phrases and information.
Response: Thank you for bringing this aspect to our attention. We have removed redundant phrases with the help of your nuanced comments.
Parallelism: Use parallel sentence structure when comparing or listing things.
Response: see below
Examples:
"The vegetation structure in the SRCs consisted of the variables ‘herbage’, ‘grass’, ‘litter’ and ‘open’ (Figure 3). Additionally, these plots were notably characterized by predominantly high shading levels (Shade). Considerable differences in the coverage levels of these variables were characteristic in the SRCs across the years, whereas this was observed to a lesser extent in the reference plots (Tables A7 and A8)." (Combine the two sentences and rephrase for clarity.)
"Across the entire study period, the ground beetle cenoses in the SRCs, except for SRC4, exhibited statistically significantly higher Shannon indices for Species-Diversity compared to the reference plots (Figure 6a)." (Simplify sentence structure and word choice.)
Response: We absolutely agree with this comment. The three sentences mentioned have been reworded and summarized (lines 325-329).
Word choice:
Some words are not used correctly or could be replaced with more precise alternatives.
Use less jargon and technical terms, especially when writing for a general audience.
Examples:
"Cenoses" instead of "communities" might be confusing for a general audience.
"Considerable differences" could be replaced with a more specific term like "substantial variations" or "marked fluctuations."
Response: We have reworded and simplified the sentence in lines 411-413 and replaced terms such as "cenoses" with communities.
Clarity and conciseness:
The writing can be more concise and impactful. Avoid unnecessary jargon and technical terms.
Ensure smooth flow between sentences and paragraphs.
Examples:
"In total, between 2011 and 2014, across all study plots, 103 ground beetle species with 46,617 individuals and 181 spider species with 42,252 individuals (167 spider species with 35,623 individuals and 14 harvestman species with 6,629 individuals) were recorded (Tables A1 and A2)." (This sentence can be broken down into several shorter sentences for better clarity.)
Response: You are absolutely right. This wording has been shortened and split into two sentences (lines 349-351).
Discussion
Grammatical and stylistic issues:
Sentence structure: Sentences are often long and complex, making them difficult to read. Aim for shorter, clearer sentences.
Response: Thank you for drawing our attention to this aspect. We have shortened the cumbersome sentence constructions with the aim of making them easier to understand. This hint is also very useful for us in the future, we need to pay more attention to this.
Word choice: Some words are not used correctly or could be replaced with more precise alternatives. Avoid jargon and technical terms for a wider audience.
Response: As mentioned above, we used an English-language editing service to improve readability. The points you kindly drew our attention to in the discussion were a great help to us after editing. Without your comments and suggestions for improvement, we would not have been able to recognize and remove them.
Redundancy: Eliminate unnecessary phrases and redundancies.
Response: Your comments not only enabled us to shorten the discussion, but also the manuscript as a whole and make it more concise.
Conciseness: The writing can be more concise and impactful. Avoid padding and overly technical terminology.
Response: Where possible, we were happy to implement this advice.
Clarity and flow: Some transitions between sentences and paragraphs could be smoother. Ensure consistent clarity and logical flow throughout the discussion.
Response: We hope that we have been able to formulate the transitions a little more smoothly.
Citations and references: Ensure proper formatting and consistent citation of references throughout the article.
Response: This has been carefully checked and corrected.
Scientific, technical, and article writing principles:
Species names: Latin species names are missing for many animal groups mentioned.
Response: A missing description of taxa has been added in line 700.
Statistical terminology: The use of some statistical terms (e.g., "highly significant") could be replaced with more specific p-values or confidence intervals.
Response: In lines 730, 731 and 784, these terms have been shortened or supplemented by the corresponding p-values.
Visual aids: Consider incorporating figures or tables to present data more effectively.
Response: We have deliberately omitted additional figures in the discussion to ensure a clear distinction from the results section.
Discussion depth: Some statements need further elaboration or supporting evidence.
Response: We could not identify any deficits with regard to this comment.
Novelty and significance: The novelty and potential impact of the findings could be emphasized more.
Response: An explanation has been added in lines 689-690. Finally, we have added a paragraph on novelty and significance in lines 827-834.
Conclusion: While the discussion touches on ecological implications, a stronger and more concise conclusion summarizing the key takeaways would be beneficial.
Response: We hope to have achieved a more concise conclusion by adding a final classification of the results within the discussion in lines 827-834 and by adding the conclusions in lines 838-857.
Specific examples of grammatical errors
In addition to the general grammatical issues mentioned above, there are a number of specific grammatical errors in the discussion section. For example:
The word "than" is used incorrectly in the following sentence:
Original: The mean proportions of arable and grassland species in both animal groups exhibit statistically significantly lower values in the oldest growth year compared to the youngest.
Revised: The mean proportions of arable and grassland species in both animal groups are statistically significantly lower in the oldest growth year than in the youngest.
Response: Thank you for your attentive review. This and the following formulations were retained even after English editing. We have adopted your suggestion and made an effort to reformulate the second part of the sentence in a grammatically correct way (lines 772-776).
The word "particularly" is used incorrectly in the following sentence:
Original: The Species- and Habitat-Preference Diversity within the SRCs is statistically significantly higher in both animal groups compared to reference plots. Particularly in highly dynamic habitats like SRC, the presence of species with various habitat preferences can be advantageous.
Revised: The Species- and Habitat-Preference Diversity within the SRCs is statistically significantly higher in both animal groups compared to reference plots. In highly dynamic habitats like SRC, the presence of species with various habitat preferences can be advantageous.
Response: In line with your suggestion, we have reworded this sentence in line 796.
The word "insurmountable" is used incorrectly in the following sentence:
Original: This suggests the potential of rotationally harvested SRC as a connecting element between wooded habitats in agricultural landscapes within surmountable distances.
Revised: This suggests the potential of rotationally harvested SRC as a connecting element between wooded habitats in agricultural landscapes within reasonable distances.
Response: We have adopted your suggestion for rewording in line 769.
We would like to thank you once again for your very detailed comments, which we have gladly taken on board.
Conclusion:
missing
Response: Thank you for drawing our attention to this. We have added the conclusions in lines 837-857.
Sincerely,
Jessika Konrad, Ralph Platen and Michael Glemnitz
Reviewer 4 Report
Comments and Suggestions for Authors
The manuscript is a good addition in ecological Entomology as well as community ecology. The study has been well planned and results inferred are of significant importance in landscape community structure and its composition. However, I have some suggestions for improvement in the attached file.

Minor revision needed
Author Response
Dear Reviewer 4,
We would like to thank you very much for your conscientious review of our manuscript. Your suggestions for improvement were very valuable and helpful to us.
In order to improve the readability of the manuscript, we have decided to use an English-language editing service.
Below you will find detailed documentation on the implementation of your individual suggestions for improvement.
Title
- The title is too long, please make it shorter. Higher taxa in the brackets like (Col.: Carabidae) should be given complete as (Coleoptera: Carabidae) for all readers.
Response: Thank you very much for your comment. You are absolutely right. We have shortened the title. However, we have refrained from publishing the higher taxa in full, as this would have made the title just as long as before despite the shortening.
Abstracts
- In first three lines title of the manuscript must be reflected in economic terms.
Response: In accordance with the journal's recommendations, the study should be placed in a broader context at the beginning. For this reason, we have made the title of the manuscript more specific as the third sentence in lines 14-16.
- Line 22 to 29 please give results in numerical terms here for quick understanding of readers.
Response: Thank you for this suggestion for improvement. These values have been added to lines 21, 24 and 26.
- Research gap of the study must be given in the last lines for future researchers.
Response: We have added the reference to an existing research gap in lines 28, 29.
Introduction
This part looks very long with respect to the title and objectives of the studies.
- According to me first three paragraphs must be summarized to a single paragraph
Response: On your advice, we have considerably shortened the introduction overall, but were unfortunately unable to reduce the first three paragraphs to a single paragraph.
- Objectives must be given in past tense as given below
The overarching objective *was* to analyse the impact of vegetation structure on the composition of ground beetle and arachnid communities within SRC in comparison to the reference plots. Please change all objectives as I have given example.
- Grammar of last paragraph needs the attention of authors.
Response: Thank you for pointing out this important point. We have corrected the tense in lines 106-125.
Materials and Methods
- Line 141 brown soils, which have a medium yield potential of 29 ??????.
Response: We are pleased that you drew our attention to the missing unit. The unit has been added in line 140.
- Line 161-182 need proper citations of methodology followed.
Response: Unfortunately, this note is incomprehensible. These lines contain a location description of the study areas.
Results
- From my side this part has been written with expertise, all results were organized excellently however from here results with numerical values must be shifted to abstract section for more understanding of common readers.
Response: This is a very convincing hint. The p-values have been added to the abstract. Thank you again for this hint.
Discussion
I think this part need a lot of improvement as many sections of results have not been discussed separately for example information given in table 1.
Response: In our opinion, this would not enrich the discussion and would also greatly expand it. A differentiated description and classification of the individual results is provided in the results section. In the discussion, we focused on classifying the results in comparison with earlier studies on ground beetles and arachnids in SRCs and on interpreting the results in a broader context. After careful consideration, we have therefore decided against this and hope for your understanding.
Conclusion
This part is missing.
Response: Thank you for bringing this to our attention. We have added the conclusions in lines 837-857.
Sincerely,
Jessika Konrad, Ralph Platen and Michael Glemnitz
Round 2
Reviewer 2 Report
Comments and Suggestions for Authors
Well done
Comments on the Quality of English LanguageMore o less
Reviewer 3 Report
Comments and Suggestions for Authors
No comment